# Control Deficits and Compensatory Mechanisms in Individuals with Chronic Ankle Instability During Dual-Task Stair-to-Ground Transition

**DOI:** 10.3390/bioengineering12101120

**Published:** 2025-10-19

**Authors:** Yilin Zhong, Xuanzhen Cen, Xiaopan Hu, Datao Xu, Lei Tu, Monèm Jemni, Gusztáv Fekete, Dong Sun, Yang Song

**Affiliations:** 1Faculty of Sports Science, Ningbo University, Ningbo 315211, China; 2211040032@nbu.edu.cn (Y.Z.); xudatao@nbu.edu.cn (D.X.); 2311040080@nbu.edu.cn (L.T.); mjemni@faculty.carrickinstitute.com (M.J.); sundong@nbu.edu.cn (D.S.); 2Centre for Mental Health Research in Association with the University of Cambridge, Cambridge CB2 1TN, UK; 3The Carrick Institute, Cape Canaveral, FL 32920, USA; 4Department of Materials Science and Mechanical Engineering, AUDI Hungaria Faculty of Engineering, Széchenyi István University, H-9026 Győr, Hungary; fekete.gusztav@sze.hu; 5Key Laboratory of Impact and Safety Engineering, Ministry of Education, Ningbo University, Ningbo 315211, China; 6Department of Biomedical Engineering, Faculty of Engineering, The Hong Kong Polytechnic University, Hong Kong 999077, China

**Keywords:** chronic ankle instability, dual-task paradigm, stair-to-ground transition, compensatory mechanisms, control deficits

## Abstract

(1) Background: Chronic ankle instability (CAI), a common outcome of ankle sprains, involves recurrent sprains, balance deficits, and gait impairments linked to both peripheral and central neuromuscular dysfunction. Dual-task (DT) demands further aggravate postural control, especially during stair descent, a major source of fall-related injuries. Yet the biomechanical mechanisms of stair-to-ground transition in CAI under dual-task conditions remain poorly understood. (2) Methods: Sixty individuals with CAI and age- and sex-matched controls performed stair-to-ground transitions under single- and dual-task conditions. Spatiotemporal gait parameters, center of pressure (COP) metrics, ankle inversion angle, and relative joint work contributions (Ankle%, Knee%, Hip%) were obtained using 3D motion capture, a force plate, and musculoskeletal modeling. Correlation and regression analyses assessed the relationships between ankle contributions, postural stability, and proximal joint compensations. (3) Results: Compared with the controls, the CAI group demonstrated marked control deficits during the single task (ST), characterized by reduced gait speed, increased step width, elevated mediolateral COP root mean square (COP-ml RMS), and abnormal ankle inversion and joint kinematics; these impairments were exacerbated under DT conditions. Individuals with CAI exhibited a significantly reduced ankle plantarflexion moment and energy contribution (Ankle%), accompanied by compensatory increases in knee and hip contributions. Regression analyses indicated that Ankle% significantly predicted COP-ml RMS and gait speed (GS), highlighting the pivotal role of ankle function in maintaining dynamic stability. Furthermore, CAI participants adopted a “posture-first” strategy under DT, with concurrent deterioration in gait and cognitive performance, reflecting strong reliance on attentional resources. (4) Conclusions: CAI involves global control deficits, including distal insufficiency, proximal compensation, and an inefficient energy distribution, which intensify under dual-task conditions. As the ankle is central to lower-limb kinetics, its dysfunction induces widespread instability. Rehabilitation should therefore target coordinated lower-limb training and progressive dual-task integration to improve motor control and dynamic stability.

## 1. Introduction

With the promotion of national sport plans and increasing public awareness of physical exercise, physical activity has become part of daily life. However, this trend is accompanied by a higher incidence of sports-related injuries, particularly those involving the lower-limb joints [1].

The ankle, as a key weight-bearing joint essential for locomotion and balance, is especially vulnerable, with acute sprains occurring in approximately 2.15 individuals per 1000 annually [2,3,4]. A major sequela is chronic ankle instability (CAI), which develops in up to 46% of sprain cases and is clinically defined by recurrent sprains, persistent “giving way”, and self-reported dysfunction [5,6]. Crucially, CAI is no longer viewed as a purely peripheral disorder of ligament laxity. Evidence now underscores central neuromuscular impairments, including proprioceptive deficits and altered sensorimotor integration, as core components of its pathophysiology [7,8]. Neuroimaging studies corroborate this, showing reduced cortical activation in individuals with CAI, which suggests that incomplete neural adaptation underpins their persistent functional deficits [9,10].

Considering the persistent sensorimotor deficits in individuals with CAI, dual-task (DT) paradigms are particularly well suited to probing the interaction between cognitive processing and motor control. Foundational theories, such as the capacity-sharing model propose that attentional resources are limited and must be allocated across concurrent tasks [11]. By simulating real-world scenarios that require divided attention (e.g., walking while talking), DT paradigms can directly measure the attentional resources necessary for maintaining stability. In individuals with CAI, gait and postural control already demand increased conscious oversight due to impaired proprioception and altered sensorimotor integration; the additional cognitive load may exceed the available attentional capacity, resulting in reduced gait stability, increased variability, and compensatory strategies, thereby revealing latent deficits [12,13,14,15]. Recent empirical studies support this notion, demonstrating that DT conditions significantly increase gait variability, postural sway, and compensatory joint strategies compared to single-task conditions [11,16,17,18]. Over the past five years, a growing body of evidence has consistently confirmed that DT paradigms effectively expose central control deficits in CAI during complex gait tasks [19,20]. Consequently, the DT paradigm serves as a critical methodological bridge for quantifying the central contributions to functional instability. Meanwhile, increasing academic attention has focused on gait cycle instability, elevated postural fluctuations, and reduced neuromuscular drive efficiency in CAI individuals under DT conditions [21,22]. Furthermore, existing studies have demonstrated that DT training can effectively enhance postural control and dynamic balance in CAI patients and even promote cortical neuroplasticity and optimization of motor coordination mechanisms [23,24,25]. We specifically selected an arithmetic task as the cognitive load because it represents a common, ecologically valid attentional demand in daily life (e.g., talking while walking, mental calculation while moving). Such a task reliably engages working memory and central executive function, which are critical for allocating attention to motor control. Many previous studies have adopted arithmetic tasks in dual-task paradigms because they reliably engage working memory and central executive functions, which are essential for attention allocation to motor control [19,26,27]. This makes it particularly suitable for revealing central control deficits in CAI, which are otherwise masked under single-task conditions.

Many deficits become evident during dynamic actions such as stair descent and the transition from stairs to ground, highlighting the greater postural control demands of these tasks [28]. But stair descent is a common daily activity with a high fall risk, placing greater demands on lower-limb muscle control and central nervous system integration [29,30]. Research indicates that 75% of stair-related fall incidents occur during the descent, particularly during the transition from stairs to flat ground, where a more significant decline in movement stability is observed [31,32,33]. Although some studies have focused on the stair descent gait characteristics of CAI patients, such as changes in ankle joint moments during the support phase and hip–knee compensation mechanisms [34,35], research on the critical “stair-to-ground transition” phase remains relatively limited. Recent studies further show that CAI patients performing dual-task activities during landing or stair descent demonstrate altered ankle inversion angles, increased angular velocity, and higher injury risk, highlighting the importance of examining this transition phase under cognitive load [20,36]. At this stage, the supporting leg must precisely control the redirection of the body’s center of mass, requiring greater central nervous system integration, especially under dual-task loads, making functional deficits more likely to appear [37]. Moreover, dual-task training can improve both static and dynamic postural stability in CAI, suggesting cognitive–motor interventions may enhance gait control and neuromuscular coordination [19,25].

In summary, this study introduced an arithmetic cognitive task during the stair-to-ground transition phase, using a dual-task paradigm to comprehensively examine gait stability, postural control, and joint compensatory patterns in individuals with CAI under concurrent cognitive and motor demands, thereby quantifying central control deficits. Combining the dual-task paradigm with the stair-to-ground transition represents a substantial challenge for individuals with chronic ankle instability who already exhibit neuromuscular impairments. Compared to single-task or level-ground gait studies, this design is not only more ecologically valid but also scientifically justified, directly allowing for the investigation of how dual-task interference affects motor performance deficits and compensatory mechanisms in CAI.

The purpose of the study was to objectively examine the deficits of control and compensatory movements of CAI during dual-task transitions between stairs and the ground. Specifically, the objectives are to (a) quantify differences in balance and stability between individuals with CAI and healthy controls using measures such as gait speed (GS) and step width (SW) and center of pressure (COP) parameters, in combination with dual-task cost (DTC); (b) to determine joint work contributions via inverse dynamics, together with joint angles, range of motion (ROM), joint moments, and statistical parametric mapping (SPM), thereby revealing the patterns of energy redistribution and compensatory strategies in CAI; (c) to examine the correlations between ankle contribution (Ankle%) and COP, spatiotemporal gait measures, and proximal joint contributions, as well as the relationships between DTC indices, in order to elucidate the role of ankle deficits in overall postural control and highlight CAI’s sensitivity to dual-task interference; and (d) to integrate multidimensional metrics to identify the control deficits and compensatory mechanisms of CAI under single- and dual-task conditions, providing quantitative evidence for rehabilitation and targeted intervention strategies.

Based on the interplay between central resource allocation and motor performance, we hypothesize the following: (a) Compared to healthy controls (CON), the CAI group will demonstrate reduced gait speed, increased step width, and greater mediolateral COP displacement, with these deficits exacerbated under a DT. (b) The CAI group will exhibit reduced ankle plantarflexion moments and work contribution (Ankle%), with compensatory increases in knee and hip contributions (Knee%, Hip%). (c) A significant negative correlation will exist between Ankle% and mediolateral COP displacement in the CAI group, underscoring the ankle’s pivotal role in dynamic stability. (d) The participants with CAI will adopt a “posture-first” strategy under the DT, manifesting as a greater dual-task cost on cognitive performance compared to that for CON.

## 2. Materials and Methods

### 2.1. Participants

A total of 60 male participants were recruited, including 30 individuals with chronic ankle instability (CAI group: age = 22.23 ± 1.72 years; weight = 77.96 ± 8.51 kg; height = 177.63 ± 6.31 cm) and 30 healthy controls (CON group: age = 22.70 ± 1.58 years; weight = 78.30 ± 8.89 kg; height = 178.17 ± 6.29 cm). Leg dominance was determined by asking the participants which leg they would use to kick a ball, and all were identified as right-leg-dominant. The inclusion criteria for the CAI group required (1) an independent stair walking ability; (2) no history of neurological disorders; (3) no previous lower-limb injuries other than right ankle sprains; (4) a history of multiple ankle sprains, with the initial sprain classified as severe (brace use >1 week or immobilization ≥3 days; (5) self-reported functional impairment (Cumberland Ankle Instability Tool score <28); and (6) the most recent sprain occurring at least six weeks prior to testing. The study protocol was approved by the Ethics Committee of Ningbo University (Approval Code: TY202455), and written informed consent was obtained from all participants.

### 2.2. Protocol

Participants wore standardized tight-fitting attire to reduce marker occlusion. Stature and body mass were recorded, followed by a 10 min warm-up (light jogging, practice stair descent, and arithmetic rehearsal). Thirty-eight retroreflective markers were placed according to the OpenSim 2392 model, as shown in Figure 1A. A static calibration trial was collected in the anatomical position to scale the model and serve as a baseline for dynamic analyses. Before stair descent testing, the participants performed a seated ST arithmetic task involving serial subtraction of three from a randomly presented three-digit number. Subsequently, each participant completed both ST and DT stair descents combined with the arithmetic task, as shown in Figure 1B, and the red ellipses indicate the stance phase of the stair-to-ground movement, which was the focus of the biomechanical analysis in this study. A three-step staircase was built in accordance with national residential standards (step height: 17 cm; tread depth: 29 cm; total height: 0.51 m; slope ≈ 30°), as shown in Figure 1B, representing a typical stair-to-ground transition. In the ST condition, the participants descended the stairs at a natural pace, leading with the right foot and landing on a force platform before continuing forward walking to a 1.5 m mark [13]. In the DT condition, the same procedure was performed concurrently with the arithmetic task. Kinematic and kinetic data were recorded throughout. Each participant performed five valid trials per condition, with a 30 s rest between trials and a 5 min rest between conditions [38]. The order of the conditions was randomized, and trials were repeated if force platform contact failed or marked gait deviations occurred. A safety observer monitored all sessions to prevent falls.

### 2.3. Data Collection and Processing

As shown in Figure 1C, motion data were acquired in the laboratory using a Vicon 3D motion capture system (Oxford Metrics Ltd., Oxford, UK; 200 Hz) synchronized with an AMTI force platform (Advanced Mechanical Technology Inc., Watertown, MA, USA; 1000 Hz). Marker trajectories were processed in Vicon Nexus1.8.5 (Oxford Metrics Ltd., Oxford, UK) for labeling, denoising, gap-filling, and smoothing. Missing marker trajectories were handled using a cubic spline interpolation method in Vicon Nexus. Gaps shorter than 10 consecutive frames were interpolated directly. For gaps longer than 10 frames, trajectories were reconstructed based on adjacent marker data and segment kinematics to ensure continuity [39]. This process utilizes algorithms such as Woltring (Quintic spline), Pattern, and Rigid Body, depending on the nature of the missing data. For gaps with sufficient surrounding valid frames, the Woltring algorithm was applied to generating a quintic spline interpolation. For gaps lacking adequate surrounding data, the Pattern or Rigid Body method was used, where the Pattern method interpolates based on a donor trajectory, and the Rigid Body method assumes the movement of the trajectory as part of a rigid body, filling the gaps accordingly. The stance phase of the transition step was defined from initial contact (vGRF > 20 N) to toe-off (vGRF < 20 N). Processed C3D files were converted in MATLAB R2023a (The MathWorks, Natick, MA, USA) and imported into OpenSim 4.4 (SimTK, Stanford, CA, USA). The Gait 2392 musculoskeletal model [40] was scaled to individual anthropometrics, and inverse kinematics (IK) and inverse dynamics (ID) were performed to compute the joint angles and moments. To ensure scaling accuracy, the model was visually inspected in OpenSim to verify marker alignment at all anatomical landmarks after scaling. Joint center locations were compared between the scaled model and experimental marker trajectories (<5 mm deviation). In addition, inverse kinematics residuals were checked to remain below 2° for all joints, confirming acceptable kinematic fidelity for subsequent joint work estimation. Furthermore, to ensure the dynamic consistency crucial for joint work calculation, a residual analysis was conducted. The normalized root mean square (RMS) of the residual forces and moments was verified to be within acceptable limits (<20 N and <30 Nm, respectively) throughout the simulation. Kinematic and kinetic data were subsequently filtered (Butterworth, 20 Hz and 50 Hz, respectively) and time-normalized to 101 points per trial for analysis [29]. Spatiotemporal parameters (gait speed, step width) and the arithmetic task score (ATS) were computed. The ATS was used to evaluate cognitive performance during single-task (ST) and dual-task (DT) conditions. In the ST condition, the ATS was calculated as the number of correct responses during a 10 s seated trial, divided by 10 to obtain the number of correct responses per second. In the DT condition, the ATS was normalized by the walking duration, with the calculation period defined from the auditory cue “start” until the participant had walked 1.5 m beyond the last step of the stair descent. This yielded the number of correct responses per second during the dual-task walking trial, ensuring comparability between the ST and DT conditions [41]. Postural stability was evaluated using center of pressure (COP) metrics: (1) total medial–lateral COP displacement (COP-ml), calculated as the cumulative sum of successive sample-to-sample differences across the gait cycle, and (2) medial–lateral COP variability, expressed as the root mean square (COP-ml RMS), representing the standard deviation of the COP relative to its mean position.(1)$$\mathrm{CO}P_{\mathrm{ml}}=\sum_{I=2}^{N}\left|\mathrm{CO}P_{\mathrm{mi}}-\mathrm{CO}P_{\mathrm{ml},i-1}\right|$$(2)$$\mathrm{CO}P_{\mathrm{ml}}^{\mathrm{RMS}}=\sqrt{\frac{1}{N}\sum_{\mathrm{in}=1}^{N}{\left(\mathrm{CO}P_{\mathrm{milli}}-{\bar{\mathrm{COP}}}_{\mathrm{ml}}\right)}^{2}}$$

Kinematic variables included the sagittal-plane ranges of hip, knee, and ankle motion, as well as the frontal-plane angles of ankle inversion (INV), knee varus, and hip abduction (ABD). Kinetic parameters comprised maximum hip flexion and extension moments (Hip FM, Hip EM), the first and second peak knee extension moments (KE M1, KE M2), and the first and second peak ankle plantarflexion moments (PF M1, PF M2), along with joint work and power.(3)$$P_{a}=M_{I}\times \omega_{a}$$(4)$$W_{\mathrm{pos}}=\int_{t1}^{t2}P\mathrm{tap}>0$$(5)$$W_{\mathrm{neg}}=\int_{t1}^{t2}P\mathrm{top}<0$$(6)$$W_{\mathrm{net}}=W_{\mathrm{pos}}+W_{\mathrm{neg}}$$(7)$$\mathrm{Con}t_{\mathrm{joint}}=\frac{\left|{W\;}_{\mathrm{joint}}^{\mathrm{net}}\right|}{\left|{W\;}_{\mathrm{Hip}}^{\mathrm{net}}\right|+\left|W_{\;\mathrm{Knee}}^{\;\mathrm{net}}\right|+\left|W_{\;\mathrm{Ankle}}^{\;\mathrm{net}}\right|}\times 100\%$$

To evaluate the effect of cognitive interference on motor performance, the DTC was calculated for the primary outcome.(8)$$\mathrm{DTC}\left(\%\right)=\frac{\left|\mathrm{DT}-\mathrm{ST}\right|}{\mathrm{ST}}\times 100$$

Finally, to investigate differences across the entire joint motion cycle, statistical parametric mapping (SPM1d) was applied. Angle and moment trajectories were normalized to the full gait cycle and analyzed with a two-factor design (group: CAI vs. CON; task: ST vs. DT). Significance was assessed using random field theory, with α set at 0.05.

### 2.4. Statistical Analysis

All statistical analyses were performed using SPSS 26.0 and Python 3.9 For discrete biomechanical outcomes, parametric assumptions were systematically evaluated: normality was tested using the Shapiro–Wilk test, and homogeneity of variance was tested using Levene’s test. For the mixed-design repeated-measures ANOVA, when both assumptions were satisfied, parametric testing was conducted, with the F-values, *p*-values, and partial η^2^ reported (0.01 = small; 0.06 = medium; 0.14 = large). When the data violated these assumptions (*p* < 0.05), a rank-transformed non-parametric mixed-design ANOVA was performed, and the F-values, *p*-values, and partial η^2^ (rank-based) were reported. For all simple-effects analyses following a significant interaction, Bonferroni correction was applied (α = 0.0125 for four pairwise comparisons). Effect sizes were reported as Cohen’s d for parametric pairwise comparisons and r for non-parametric analyses (0.1 = small, 0.3 = medium, 0.5 = large). In the dual-task cost (DTC) analysis, the normality (the Shapiro–Wilk test) and homogeneity of variance (Levene’s test) of all variables were first assessed. According to the data characteristics, independent-samples *t*-tests were applied when the data were approximately normal with equal variances, Welch’s *t*-tests were applied when the variances were unequal, and Mann–Whitney U tests were applied when normality was not satisfied or only partially met.

For kinematic and kinetic time-series data, statistical parametric mapping (SPM) was applied to the ankle, knee, and hip joint angles and moments. Within-group comparisons (single-task vs. dual-task) used paired *t*-tests, and between-group comparisons (CAI vs. CON) used independent-sample *t*-tests, all performed in the SPM framework. Time-series data were normalized to the stance phase (0–100%), and the family-wise error rate across the time series was controlled using random field theory (RFT) correction. Significant time regions are reported with their time intervals (% gait cycle), peak t-values, corrected *p*-values, and descriptive information from mean trajectories.

Pearson’s correlation and linear regression analyses were conducted to examine the relationships among joint contributions (Ankle%, Knee%, Hip%), COP-ml RMS, gait speed, step width, ankle inversion kinematics, and dual-task costs (DTC Ankle%, DTC COP-ml RMS). Correlation coefficients (r), regression coefficients (β), coefficients of determination (R^2^), 95% confidence intervals, and *p*-values are reported, with the significance set at *p* < 0.05. The residuals from all regression models were visually inspected and tested for normality.

## 3. Results

### 3.1. Balance and Stability

Normality was tested using the Shapiro–Wilk test and homogeneity of variance using Levene’s test. When both assumptions were satisfied, a parametric mixed-design repeated-measures ANOVA was conducted, with the F-values, *p*-values, and partial η^2^ reported. When the data violated these assumptions (*p* < 0.05), a rank-transformed non-parametric mixed-design ANOVA was performed, and the F-values, *p*-values, and partial η^2^ (rank-based) are reported in Table 1. For all simple-effects analyses following a significant interaction, Bonferroni correction was applied (α = 0.0125 for four pairwise comparisons). The detailed normality and homogeneity tests for all variables are provided in Table A1 and Table A2.

The center of pressure mediolateral root mean square (COP-ml RMS) was analyzed using non-parametric methods due to the severe normality violation in the CAI group under DT conditions. Significant task and interaction effects were observed, with no significant group effect. The parametric ANOVA yielded consistent results. Simple effect analyses showed that in the CON group, COP-ml RMS was higher under the DT than the ST (w = 14.00, z = 4.49, *p* < 0.001, r = 0.58), while the group with CAI showed a significant difference (w = 24.00, z = 4.28, *p* < 0.001, r = 0.55). No group differences were found under either task, indicating that the DT increased postural sway in both groups. COP mediolateral excursion (COP-ml) was analyzed non-parametrically due to moderate heterogeneity under DT conditions. Significant task and interaction effects were found, with no group effect. The parametric ANOVA results were consistent. Simple effects showed that the DT increased COP-ml in both the CAI (w = 118.00, z = 2.36, *p* = 0.02, r = 0.30) and CON (w = 5.00, z = 4.68, *p* < 0.001, r = 0.60) groups, with no significant group differences, suggesting a consistent DT-induced lateral displacement in the COP across groups. Step width (SW) was analyzed non-parametrically due to severe normality violations. The task effect was significant, whereas group and interaction effects were not. The parametric ANOVA confirmed these results, showing that the DT increased SW in both groups. Gait speed (GS) was analyzed using non-parametric methods due to moderate violations of parametric assumptions. Significant task and interaction effects were observed, with no group effect. Simple effects indicated that GS decreased under the DT compared with that under the ST in CAI (W = 0.00, Z = 4.78, *p* < 0.001, r = 0.62), whereas CON showed no significant difference. No group differences were found under either task, suggesting that the DT specifically impaired GS in the CAI patients. Arithmetic task score (ATS) was analyzed parametrically. Significant group and interaction effects were found, with no task effect. Simple effect analyses revealed that the ATS decreased under the DT compared with that under the ST in CAI (t(29) = 5.42, *p* < 0.001, d = 1.47), whereas it increased in CON (t(29) = −3.95, *p* < 0.001, |d| = 0.85). Under DT conditions, the CAI group scores were significantly lower than those in CON (t(58) = −8.48, *p* < 0.001, |d| = 2.19), with no significant group difference under the ST (t(58) = −0.30, *p* = 0.77, d = 0.08). These results suggest that the DT selectively impaired arithmetic performance in the CAI patients while enhancing it in the healthy controls.

### 3.2. Joint Kinematics and Kinetics

Normality was tested using the Shapiro–Wilk test and homogeneity of variance using Levene’s test. When both assumptions were satisfied, parametric mixed-design repeated-measures ANOVA was conducted, with F-values, *p*-values, and partial η^2^ reported. When the data violated these assumptions (*p* < 0.05), a rank-transformed non-parametric mixed-design ANOVA was performed, and F-values, *p*-values, and partial η^2^ (rank-based) were reported in Table 2. The detailed normality and homogeneity tests for all variables are provided in Table A1 and Table A2.

Significant differences in lower-limb joint range of motion (ROM) were observed in both the sagittal and frontal planes. For ankle ROM, severe heterogeneity under DT conditions warranted non-parametric analyses. Group and task main effects were significant, with no interaction. The CAI group exhibited a greater ankle ROM than that in CON, and the ROM in the DT was increased relative to that under the ST. Knee ROM met the parametric assumptions, showing a significant group effect without task or interaction effects. CAI shows a greater knee ROM than that in CON. Hip ROM required non-parametric analyses due to normality violations in CON; group, task, and interaction effects were significant. Simple effects indicated a reduced hip ROM under the DT in CAI (W = 0.001, Z = 4.78, *p* < 0.001 r = 0.62), no task difference in CON, and a lower ROM for CAI than CON under the DT (U = 69.00, Z = 5.63, *p* < 0.001, r = 0.73) but not the ST, suggesting a DT-specific reduction in CAI. In the frontal plane, ankle inversion showed significant group, task, and interaction effects, with the inversion in CAI greater under the DT than the ST (W = 0.001, Z = 4.78, *p* < 0.001, r = 0.62) and higher than that in CON in both the ST (U = 847.00, Z = 5.87, *p* = 0.001, r = 0.76) and the DT (U = 893.00, Z = 6.55, *p* = 0.001, r = 0.85). Knee varus exhibited a significant task and interaction effect; CAI showed increased varus under the DT (W = 95.00, Z = 2.83, *p* = 0.004, r = 0.37), with no group differences. Hip abduction showed a significant group effect, with greater abduction in CAI than that in CON.

For the joint moments, Hip FM required non-parametric analyses due to normality/variance violations; task and interaction were significant, with simple effects showing that the DT increased Hip FM only in CON (W = 13.00, Z = 4.52, *p* < 0.001, r = 0.58) and no group differences. Hip EM met the parametric assumptions; task and interaction were significant, showing a decreased EM under the DT in both groups, with no group differences. KE M1 showed task and interaction effects, with a higher KE M1 under the DT for CAI (W = 3.00, Z = 4.72, *p* < 0.001, r = 0.61) and CON (W = 7.00, Z = 4.64, *p* < 0.001, r = 0.60), with no group differences. KE M2 showed only a task effect, which was higher under the ST. PF M1 showed significant group, task, and interaction effects; PF M1 in CAI decreased under the DT (W = 1.00, Z = 4.76, *p* < 0.001, r = −0.62) and was lower than that in CON under the DT (U = 162.00, Z = 4.26, *p* = 0.001, r = 0.55), with no significant ST difference. PF M2 showed group and interaction effects; in CON, PF M2 increased under the DT (W = 50.00, Z = 3.75, *p* < 0.001, r = 0.49), while in CAI, it remained lower under both conditions (ST: U = 148.00, Z = 4.47, *p* < 0.001, r = 0.58; DT: U = 58.00, Z = 5.80, *p* < 0.001, r = 0.75).

The results of the comparison of lower-limb joint angle and moment in sagittal plane during stair-to-ground tran-sition both across task (ST vs. DT) and group (CAI vs. CON) conditions using SPM are shown in Figure 2.

The results of the joint contribution are shown in Figure 3 and Table 2, and the detailed normality and homogeneity tests for all variables are provided in Table A1 and Table A2. hip contribution was analyzed parametrically, as the data met the assumptions. The results revealed significant main effects of group and task, with a non-significant interaction. These findings indicate that hip contribution was higher in CAI than that in CON, and DT conditions consistently increased the hip contribution compared with that in the ST. Knee contribution was analyzed non-parametrically due to violations of normality and homogeneity of variance. The results showed significant main effects of group and task, with no significant interaction. The parametric ANOVA yielded consistent conclusions. These results indicate that knee contribution was higher in CAI than that in CON, and DT conditions consistently increased the knee contribution across groups. Ankle contribution was analyzed non-parametrically due to severe normality violation and moderate heterogeneity of variance. Significant main effects of group and task were observed, with a non-significant interaction. The parametric ANOVA results were consistent. These findings indicate that ankle contribution was lower in CAI than that in CON, and DT conditions consistently the reduced ankle contribution compared with that in the ST.

### 3.3. Predictive Relationships Between Ankle% and Lower-Limb Biomechanics

Correlation analyses between Ankle%, COP-ml RMS, dual-task cost (DTC), and other lower-limb biomechanical variables are shown in Figure 4. Several significant associations were identified. In the CAI group, Ankle% was significantly and negatively correlated with the COP-ml RMS under both ST and DT conditions. Regression analysis confirmed that Ankle% was a significant predictor of COP-ml RMS (ST: β = −0.28, R^2^ = 0.96; DT: β = −0.50, R^2^ = 0.47), whereas no significant associations were observed in the CON group. Additionally, Ankle% was positively correlated with gait speed (GS) under the DT in the CAI group, with the regression analysis demonstrating significant predictive power (β = 0.03, R^2^ = 0.53). In contrast, in the CON group, Ankle% showed significant positive correlations with GS under both ST and DT conditions (ST: β = 0.02, R^2^ = 0.61; DT: β = 0.02, R^2^ = 0.26). Moreover, Ankle% was strongly and negatively correlated with Knee% in both conditions for the CAI group, and regression analysis confirmed its predictive effect (ST: β = −0.87, R^2^ = 0.85; DT: β = −0.88, R^2^ = 0.82). Similar strong negative correlations were also found in the CON group (ST: β = −0.88, R^2^ = 0.84; DT: β = −0.96, R^2^ = 0.88). In addition, COP-ml RMS was positively correlated with ankle inversion (Ankle INV) in the CAI group, with regression models supporting its predictive effect (ST: β = 0.27, R^2^ = 0.43; DT: β = 0.19, R^2^ = 0.23), whereas no such relationship was evident in the CON group. COP-ml RMS was also positively correlated with ankle range of motion (ROM) in the CAI group, with significant predictive value (ST: β = 1.55, R^2^ = 0.54; DT: β = 0.63, R^2^ = 0.57), but no significance was observed in the CON group. Furthermore, DTC Ankle% was significantly and negatively correlated with DTC COP-ml RMS in both groups, with regression analyses indicating significant predictive effects (CAI: β = −0.55, R^2^ = 0.26; CON: β = −0.34, R^2^ = 0.44).

### 3.4. Dual-Task Cost

Descriptive statistics for each outcome are summarized in Table 3. Values are presented as the mean ± SD for parametric analyses and the median (Q1, Q3) for non-parametric (rank-transformed) analyses, and detailed normality and homogeneity tests for all variables are provided in Table A3 and Table A4. In the dual-task cost (DTC) analysis, the normality (the Shapiro–Wilk test) and homogeneity of variance (Levene’s test) of all variables were first assessed. According to the data characteristics, independent-samples *t*-tests were applied when the data were approximately normal with equal variances, Welch’s *t*-tests when the variances were unequal, and Mann–Whitney U tests when normality was not satisfied or only partially met. The results showed that there were no significant differences between the CAI and CON groups in ankle inversion angle and step width (SW) DTCs (U = 486.50, *p* = 0.59, r = 0.07; U = 350.00, *p* = 0.14, r = −0.19), whereas the gait speed (GS) DTC was significantly higher in the CAI group than that in the CON group (t = 6.69, *p* < 0.001, d = 1.723). COP-ml DTC was significantly higher in the CAI group (t = 7.62, *p* < 0.001, d = 4.22), while COP-ml RMS DTC showed no significant difference (U = 332.00, *p* = 0.08, r = 0.23). The ankle contribution (Ankle%) and arithmetic task score (ATS) DTCs were significantly higher in the CAI group (U = 645.00, *p* = 0.004, r = 0.37; U = 604.00, *p* = 0.02, r = 0.29), whereas the knee (Knee%) and hip (Hip%) contribution DTCs were significantly lower than in the CON group (Welch t: t = −9.11, *p* < 0.001, d = −1.66; U = 67.50, *p* < 0.001, r = −0.73; Welch t: t = −6.11, *p* < 0.001, d = −1.25; U = 132.00, *p* < 0.001, r = −0.61). PFM1 DTC showed no significant difference (U = 400.00, *p* = 0.46, r = −0.10), while PFM2 DTC was significantly higher in the CAI group (t = 3.63, *p* = 0.001, d = 2.64). The KEM1 and KEM2 DTCs were both significantly higher in the CAI group (U = 713.00, *p* = 0.001, r = 0.50; U = 623.00, *p* = 0.011, r = 0.33). The Hip FM and Hip EM DTCs showed no significant differences (U = 564.00, *p* = 0.093, r = 0.22; U = 461.50, *p* = 0.87, r = 0.02). The ankle range of motion (ANKLE ROM) DTC did not differ significantly between groups (t = −0.08, *p* = 0.94, d = −0.02), whereas the knee (KNEE ROM) and hip (HIP ROM) range of motion DTCs were significantly higher in the CAI group (U = 644.00, *p* = 0.004, r = 0.30; U = 595.00, *p* = 0.03, r = 0.28).

## 4. Discussion

This study examined the compensatory movement patterns and neuromuscular control in individuals with chronic ankle instability (CAI) during stair-to-ground transitions, focusing on the influence of dual-task (DT) demands. By integrating spatiotemporal parameters, center of pressure (COP) data, joint mechanics, and cognitive performance, distinct postural control deficits and compensatory strategies were identified. The participants with CAI exhibited diminished ankle function and increased reliance on proximal joints, especially under DT conditions, resulting in reduced movement efficiency and stability. These outcomes confirm our hypotheses: (a) a slower gait speed and greater step width, (b) reduced ankle and increased knee/hip contributions, (c) negative associations between ankle work and COP stability, and (d) cognitive decline under the DT reflecting a posture-first strategy. Collectively, the results highlight both peripheral and central mechanisms underlying CAI and suggest that dual tasking exposes hidden instability not observable during simple walking.

### 4.1. Postural Stability and Attentional Demands Under Dual-Task Conditions

The spatiotemporal, COP, and cognitive data consistently revealed impaired postural control in CAI, which further deteriorated under DT conditions. The slower gait and wider step width observed even under single-task (ST) conditions indicate a conservative gait pattern adopted to enhance stability [8,42,43]. Under DT conditions, these adaptations were amplified, and the increased dual-task cost reflected limited gait automatization and greater reliance on conscious control [11,16,44]. Greater mediolateral COP displacement and variability further confirmed compromised frontal-plane stability. Such deficits likely arise from delayed peroneal activation and altered corticospinal modulation [42,45,46]. The pronounced instability under the DT supports the hypothesis that individuals with CAI rely heavily on attentional resources to maintain balance, consistent with the capacity-sharing model [18,47,48,49]. Cognitive performance outcomes reinforce this interpretation. While the controls (CON) slightly improved under the DT, the participants with CAI exhibited marked performance declines, confirming a posture-first trade-off [14,18,50]. For CON, moderate motor demand may enhance arousal and attention; for CAI, excessive attentional resources are devoted to maintaining balance [11,41]. This difference underscores that CAI involves not only peripheral deficits but also central limitations in attentional resource allocation [10,51,52].

### 4.2. Distal Suppression and Proximal Compensation

The biomechanical results revealed a clear distal-to-proximal redistribution of the joint work in CAI. Reduced ankle contribution (Ankle%) and increased knee and hip work confirm the compensatory cascade predicted in hypothesis (b), which intensified under DT conditions. This redistribution likely reflects a protective strategy to minimize distal instability [42,53,54,55]. Early plantarflexion and persistent inversion patterns indicate altered sensorimotor control and impaired propulsion [18,45].

Increased knee flexion and hip abduction may lower the center of mass and widen the base of support, improving stability but increasing proximal joint loading [56,57]. Such compensations may help task completion yet elevate overuse risk at the knee or the hip. Importantly, the negative correlation between the dual-task cost of Ankle% and COP stability suggests that maintaining the distal contribution requires attentional resources—when divided, compensatory mechanisms weaken [34,35,58,59].

### 4.3. Regulatory Role of Ankle Function in Motor Control

The regression results confirmed the ankle’s central role in maintaining gait stability and efficiency. Reduced Ankle% correlated with larger COP-ml RMS, demonstrating that ankle mechanical output directly influences mediolateral control [42,60]. The positive relationship between Ankle% and gait speed links impaired ankle propulsion to a slower, cautious gait. The inverse correlation between Ankle% and Knee% reflects energy redistribution along the kinetic chain [34,61], which may increase proximal stress and contribute to patellofemoral pain [59,62]. Additionally, the association between COP-ml RMS and ankle inversion suggests a feedback loop: poor proprioceptive control leads to abnormal positioning, amplifying instability further [10,51]. The correlation between ankle work and dual-task cost highlights that distal control deteriorates under cognitive load, reinforcing the need for dual-task-oriented rehabilitation [22,25].

### 4.4. Clinical Implications

These findings have clear clinical implications. The amplified deficits under the DT suggest that dual-task assessments can reveal functional instability better than single-task evaluations. Confirming all four hypotheses (a–d), our results emphasize that rehabilitation should target both distal control and cognitive–motor coordination. Quantifying proximal compensations during stair tasks may also inform individualized interventions.

Rehabilitation should progress from restoring local ankle sensorimotor function to retraining coordinated ankle–knee–hip energy transfer. Incorporating DT paradigms in advanced stages—supported by evidence of DT training efficacy [23,25,63]—can enhance automaticity and prepare patients for real-world multi-tasking.

### 4.5. Limitations and Future Directions

Several limitations should be acknowledged. The sample included only young males, limiting the generalizability. The modest sample size (N = 30 per group) may have reduced the statistical robustness. Uncontrolled variables such as fatigue and limb dominance may have introduced variability. The three-step staircase restricted task duration and may have underestimated sustained compensatory adaptations. Upper-body and neurophysiological data were not recorded, preventing direct interpretation of central control mechanisms. Although the arithmetic task is widely used, population-specific validation is lacking, and potential learning effects cannot be excluded. Finally, injury severity was not stratified, which may have influenced individual responses [64].

Future research should include larger and more diverse samples, extended stair descent tasks, and multimodal approaches (e.g., EMG, EEG, fNIRS). Meanwhile, finite element analysis has been extensively applied in sports biomechanics to studying the internal biomechanical behavior of human tissues [65,66,67]. Future research should combine finite element analysis with high-resolution medical imaging to obtain individualized data for CAI and develop high-fidelity ankle models. Such progress would facilitate a more thorough exploration of internal biomechanical responses in CAI during diverse movements, thereby offering deeper insights into the mechanisms of injury.

## 5. Conclusions

In this cohort of young male participants, this study investigated neuromuscular control in individuals with chronic ankle instability (CAI) during a stair-to-ground transition, revealing key deficits that were systematically exacerbated under dual-task conditions. The findings demonstrate that CAI is characterized by a guarded gait pattern, a robust “distal-to-proximal” compensatory strategy, and a marked reliance on attentional resources for postural control, as evidenced by the adoption of a “posture-first” strategy.

Theoretically, by integrating an ecologically valid motor task with a cognitive challenge, this work provides behavioral evidence consistent with models of disrupted central sensorimotor integration and resource allocation in CAI. The observed performance deterioration under cognitive load, without direct neural measures, offers compelling indirect support for the involvement of higher-order cognitive processes in CAI pathophysiology.

Clinically, the results highlight the potential value of incorporating dual-task assessments and kinetic chain evaluations to identify functional deficits better. The proposed staged rehabilitation framework—progressing from local ankle recovery to inter-joint coordination and finally to cognitive–motor integration—offers a rational approach for future intervention studies to validate.

## Figures and Tables

**Figure 1 bioengineering-12-01120-f001:**
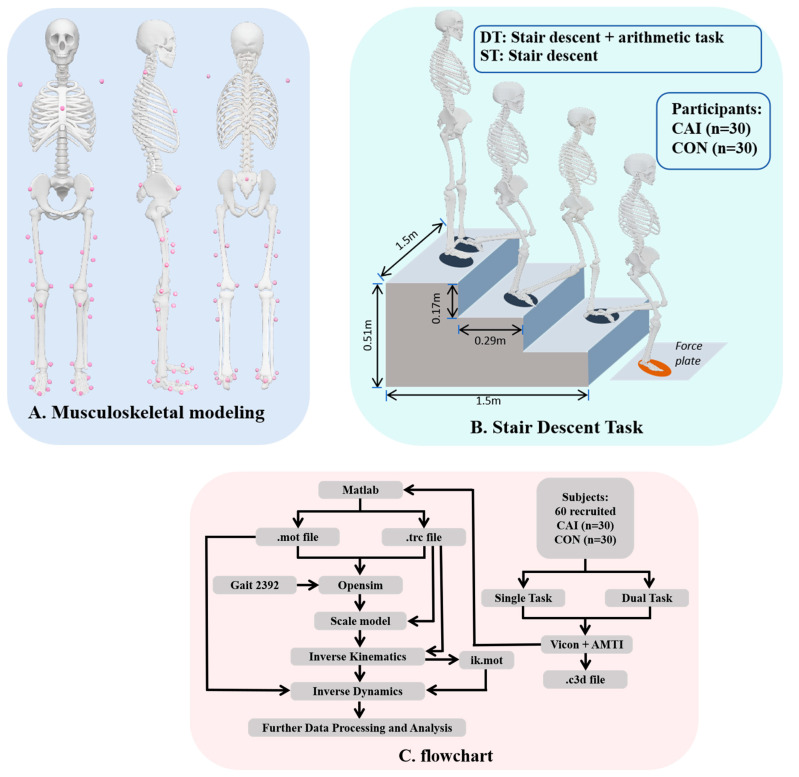
An overview of the data collection and the model. (**A**) An illustration of the reflective marks’ positions on the musculoskeletal model used in the experiment, from left to right: anterior, lateral, and posterior views. (**B**) Decomposition composition of the stair descent task, staircase dimensions, and the placement of the force platform in the laboratory. The red ellipses indicate the stance phase of the stair-to-ground movement, which is the focus of the biomechanical analysis in this study. (**C**) Flowchart of the experimental procedure and preliminary data processing. Abbreviations: CAI, chronic ankle instability; CON, control; ST, single-task; DT, dual-task.

**Figure 2 bioengineering-12-01120-f002:**
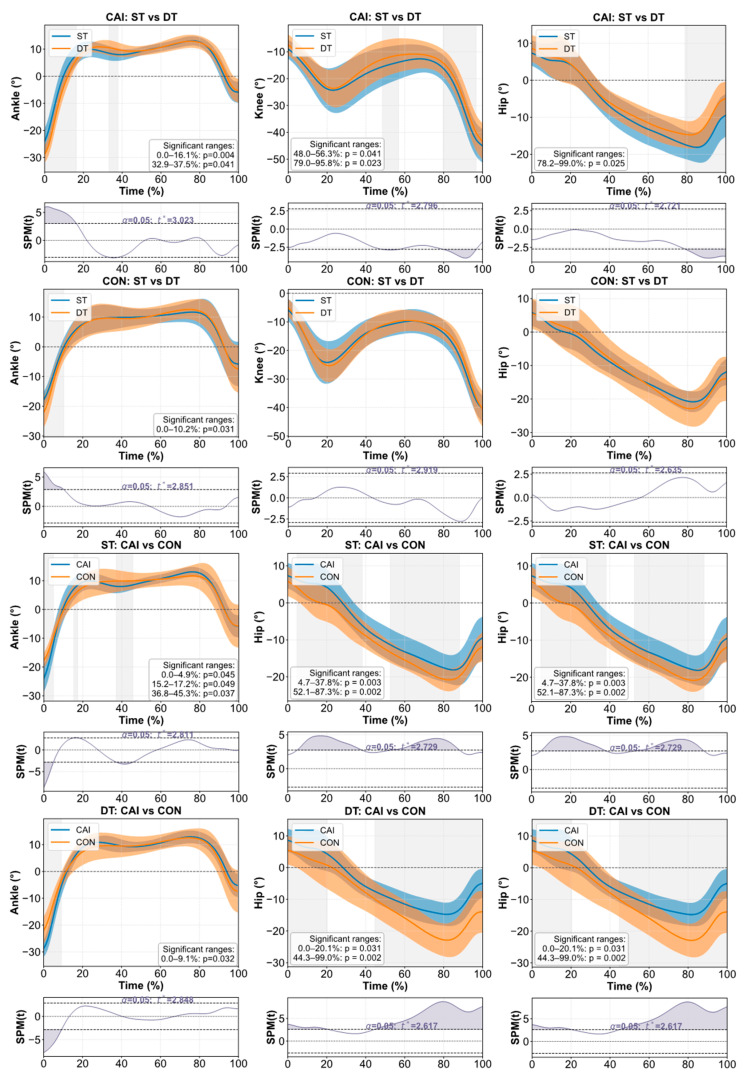
Comparison of lower-limb joint angle and moment in sagittal plane during stair-to-ground transition across task (ST vs. DT) and group (CAI vs. CON) conditions using SPM. Time-normalized kinematic and kinetic waveforms for the ankle, knee, and hip joints during the stair-to-ground transition. The solid lines represent mean trajectories, and the shaded bands represent ±1 SD. “*****” indicates a significant difference (*p* < 0.05) and gray shaded areas along the *x*-axis indicate time intervals where statistically significant differences (*p* < 0.05) were found between the CAI and CON groups under single-task (ST) conditions, the CAI and CON groups under dual-task (DT) conditions, ST and DT conditions within the CAI group, and ST and DT conditions within the CON group. Abbreviations: CAI, chronic ankle instability; CON, control; ST, single-task; DT, dual-task; SPM, statistical parametric mapping.

**Figure 3 bioengineering-12-01120-f003:**
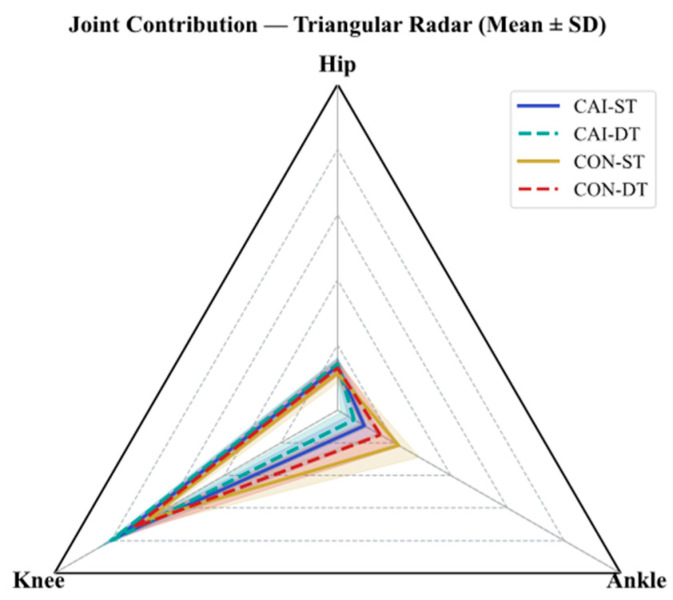
Relative joint work contributions at the ankle, knee, and hip. The stacked bar charts show the percentage contribution of each joint to the total lower-limb positive work during the stair-to-ground transition for the chronic ankle instability (CAI) and control (CON) groups under single-task (ST) and dual-task (DT) conditions. Abbreviations: CAI, chronic ankle instability; CON, control; ST, single-task; DT, dual-task; Ankle%, relative positive work contribution of the ankle joint; Knee%, knee joint; Hip%, hip joint.

**Figure 4 bioengineering-12-01120-f004:**
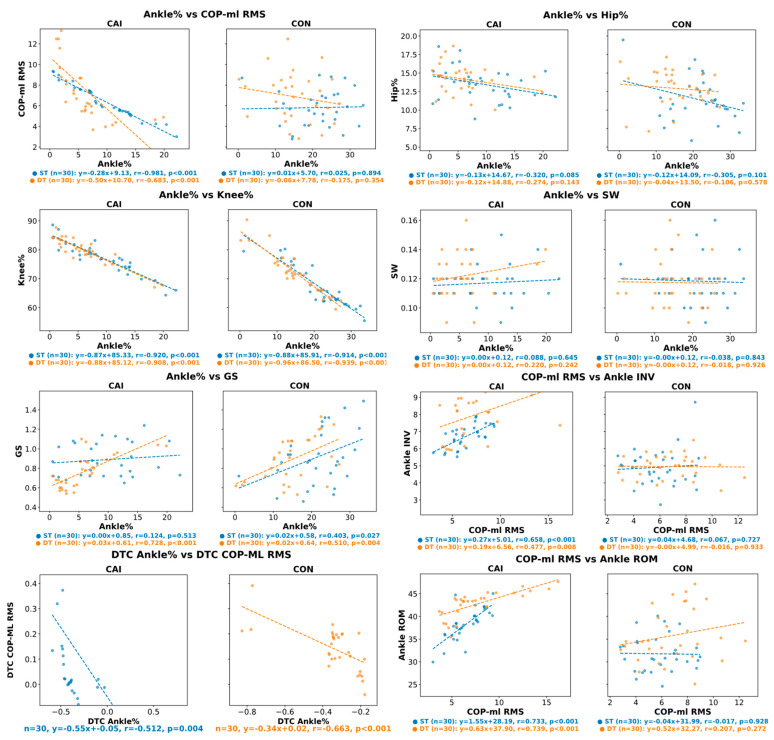
Relationships between ankle work contribution (Ankle%) and key biomechanical variables. Scatter plots with regression lines illustrate the correlations between Ankle% and the mediolateral center of pressure variability (COP-ML RMS), gait speed (GS), and knee work contribution (Knee%) for the chronic ankle instability (CAI) and control (CON) groups under both single-task (ST) and dual-task (DT) conditions. Pearson’s correlation coefficient (r) is reported for each significant relationship. Abbreviations: COP-ml RMS, mediolateral root mean square of center of pressure displacement; GS, gait speed; CAI, chronic ankle instability; CON, control.

**Table 1 bioengineering-12-01120-t001:** Comparison of center of pressure (COP) and spatiotemporal gait measures during stair-to-ground transition across task (single-task vs. dual-task) and group (CAI vs. CON) conditions.

Indicator	CAI		CON		P1	F1	η^2^1	P2	F2	η^2^2	P3	F3	η^2^3
	ST	DT	ST	DT									
ATS (point)	0.59 ± 0.05	0.51 ± 0.06	0.59 ± 0.05	0.64 ± 0.07	<0.001	40.38	0.41	0.11	2.60	0.04	<0.001	44.71	0.44
COP-ml (mm)	80.54 (73.77, 94.07)	101.31 (81.51, 109.49)	94.04 (81.59, 99.67)	96.63 (87.48, 109.92)	0.47	0.53	0.009	<0.001	75.17	0.56	<0.001	14.40	0.20
COP-ml RMS (mm)	6.15 (5.40, 7.60)	6.31 (5.48, 8.69)	5.62 (4.22, 6.91)	6.30 (4.98, 8.28)	0.26	1.31	0.02	<0.001	36.04	0.38	0.03	4.95	0.08
GS (m/s)	0.862 (0.73, 1.07)	0.74 (0.62, 0.87)	0.89 (0.76, 1.11)	0.91 (0.68, 1.08)	0.20	1.69	0.03	<0.001	168.05	0.74	<0.001	118.16	0.67
SW (m)	6.15 (5.40, 7.60)	6.31 (5.48, 8.69)	5.62 (4.22, 6.91)	6.30 (4.98, 8.28)	0.68	0.17	0.003	<0.001	139.14	0.71	0.64	0.22	0.004

Values are presented as mean ± SD for parametric analyses and median (Q1, Q3) for non-parametric (rank-transformed) analyses. P1 refers to the main effect of group (F1, p1, partial η^2^1/pseudo-η^2^1); P2 refers to the main effect of task (F2, p2, partial η^2^2/pseudo-η^2^2); P3 refers to the group × task interaction (F3, p3, partial η^2^3/pseudo-η^2^3). Abbreviations: CAI, chronic ankle instability; CON, control; ST, single-task; DT, dual-task; COP-ml, total mediolateral displacement of center of pressure; COP-ml RMS, mediolateral RMS of COP; GS, gait speed; SW, step width; ATS, arithmetic task score.

**Table 2 bioengineering-12-01120-t002:** Comparison of lower-limb joint kinematics and kinetics during stair-to-ground transition across task (single-task vs. dual-task) and group (CAI vs. CON) conditions.

Indicator	CAI		CON		P1	F1	η^2^1	P2	F2	η^2^2	P3	F3	η^2^3
	ST	DT	ST	DT									
Ankle ROM	38.41 (36.14, 40.18)	43.45 (41.37, 44.45)	30.75 (28.78, 34.01)	34.47 (32.03, 41.17)	<0.001	39.88	0.41	<0.001	334.68	0.85	0.07	3.45	0.06
Knee ROM	36.42 ± 3.80	36.92 ± 4.04	34.08 ± 4.17	35.13 ± 4.64	0.04	4.63	0.07	0.12	2.51	0.04	0.57	0.32	0.006
Hip ROM	26.50 (25.77, 27.34)	23.70 (22.52, 25.21)	26.828 (26.04, 27.97)	27.39 (25.99, 28.28)	<0.001	32.48	0.36	<0.001	13.80	0.19	<0.001	24.04	0.29
Ankle Inversion	6.80 (6.32, 7.19)	8.37 (6.96, 9.09)	4.83 (4.23, 5.44)	5.02 (4.44, 5.50)	<0.001	185.48	0.76	0.002	10.57	0.15	0.003	9.70	0.14
Knee Varus	1.15 (0.88, 1.87)	2.29 (1.98, 3.97)	2.18 (1.57, 2.80)	2.09 (1.71, 2.51)	0.23	1.45	0.02	0.004	8.92	0.13	0.01	6.72	0.10
Hip Abduction	6.42 (2.95, 7.98)	8.01 (3.29, 9.38)	4.67 (3.22, 6.80)	4.60 (2.29, 8.38)	0.03	5.04	0.08	0.52	0.42	0.007	0.37	0.84	0.01
Hip FM	0.32 (0.23, 0.36)	0.33 (0.26, 0.39)	0.34 (0.25, 0.37)	0.35 (0.27, 0.40)	0.48	0.51	0.009	<0.001	60.15	0.51	0.006	8.11	0.12
Hip EM	0.65 ± 0.10	0.51 ± 0.07	0.60 ± 0.09	0.52 ± 0.11	0.37	0.80	0.01	<0.001	100.32	0.63	0.003	9.93	0.15
KE M1	0.27 (0.15, 0.44)	0.52 (0.31, 0.65)	0.35 (0.30, 0.42)	0.43 (0.40, 0.55)	0.46	0.56	0.01	<0.001	150.90	0.72	0.002	5.93	0.009
KE M2	0.10 (0.07, 0.14)	0.08 (0.06, 0.09)	0.09 (0.08, 0.11)	0.08 (0.07, 0.10)	0.69	0.17	0.003	<0.001	116.63	0.67	0.43	0.63	0.01
PF M1	1.18 (1.15, 1.25)	1.127 (1.074, 1.191)	1.30 (1.11, 1.45)	1.256 (1.19, 1.33)	<0.001	13.80	0.19	0.002	10.96	0.16	0.004	8.85	0.13
PF M2	1.13 (1.09, 1.27)	1.04 (0.92 1.169)	1.29 (1.25, 1.41)	1.40 (1.35, 1.46)	<0.001	92.99	0.62	0.41	0.70	0.01	<0.001	17.33	0.23
Hip%	13.49 ± 2.28	14.20 ± 1.96	11.41 ± 2.98	12.94 ± 2.47	<0.001	17.15	0.23	0.02	5.30	0.08	0.41	0.69	0.01
Knee%	77.02 (74.14, 79.78)	80.13 (78.45, 83.24)	64.59 (62.39, 73.93)	71.68 (66.63, 75.21)	<0.001	74.77	0.56	0.001	11.99	0.17	0.95	0.004	7.20
Ankle%	8.53 (5.29, 13.61)	5.34 (2.30, 7.16)	22.62 (18.11, 26.30)	14.33 (11.82, 20.50)	<0.001	95.83	0.62	<0.001	15.67	0.21	0.61	0.56	0.004

Values are presented as mean ± SD for parametric analyses and median (Q1, Q3) for non-parametric (rank-transformed) analyses. P1 refers to the main effect of group (F1, p1, partial η^2^1/pseudo-η^2^1); P2 refers to the main effect of task (F2, p2, partial η^2^2/pseudo-η^2^2); P3 refers to the group × task interaction (F3, p3, partial η^2^3/pseudo-η^2^3). Abbreviations: CAI, chronic ankle instability; CON, control; ST, single-task; DT, dual-task; ROM, range of motion; Ankle%, relative ankle work contribution; Knee%, knee joint; Hip%, hip joint.

**Table 3 bioengineering-12-01120-t003:** Comparison of dual-task cost (DTC) in spatiotemporal, stability, and lower-limb biomechanical indicators between CAI and CON groups during stair-to-ground transition.

Indicator	CAI	CON
Ankle inversion DTC	23.07 (9.99, 25.61)	16.12 (9.69, 28.33)
SW DTC	10.10 (8.51, 11.05)	11.82 (8.57, 13.00)
GS DTC	14.83 ± 4.00	7.68 ± 4.27 ^a^
COP-ml DTC	17.94 ± 8.65	5.21 ± 3.02 ^a^
COP-ml RMS DTC	7.95 (1.83, 14.92)	14.36 (10.03, 19.84)
Ankle% DTC	44.05 (36.16, 49.02)	33.22 (22.53, 35.12) ^a^
Knee% DTC	1.24 (1.19, 1.26)	7.95 (4.75, 10.23) ^a^
Hip% DTC	4.995 (2.51, 9.66)	16.33 (11.92, 22.17) ^a^
ATS DTC	14.295 (10.09, 22.08)	8.05 (3.46, 20.12) ^a^
PF M1 DTC	5.435 (4.78, 6.63)	8.69 (2.12, 13.09)
PF M2 DTC	15.37 ± 10.92	7.90 ± 2.83 ^a^
KE M1 DTC	78.02 (48.09, 95.07)	24.19 (17.15, 33.50) ^a^
KE M2 DTC	21.01 (13.67, 29.97)	13.10 (11.26, 14.61) ^a^
Hip FM DTC	8.44 (3.74, 11.60)	6.28 (2.35, 8.34)
Hip EM DTC	22.18 (20.04, 23.45)	22.23 (18.98, 23.33)
Ankle ROM DTC	12.17 ± 3.43	12.26 ± 5.50
Knee ROM DTC	5.62 (3.57, 14.32)	2.63 (1.36, 5.03) ^a^
Hip ROM DTC	9.94 (8.16, 12.45)	7.45 (2.81, 11.37) ^a^

Values are presented as mean ± SD for parametric analyses and median (Q1, Q3) for non-parametric (rank-transformed) analyses. “a” indicates a significant difference between CAI and CON groups (*p* < 0.05). Abbreviations: CAI, chronic ankle instability; CON, control; DTC, dual-task cost; COP, center of pressure; ATS, arithmetic task score; GS, gait speed. SW, step width; PF M1, First peak ankle plantarflexion moment; KE M1, First peak knee extension moment; FM, flexion moment; EM, extension moment; ROM, rang of motion.

## Data Availability

The data presented in this study are available on request from the corresponding author. The data are not publicly available due to ethical considerations.

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
