# Peer review of "Control Deficits and Compensatory Mechanisms in Individuals with Chronic Ankle Instability During Dual-Task Stair-to-Ground Transition"

_bioengineering, 2025, doi:10.3390/bioengineering12101120_

Round 1

Reviewer 1 Report

Comments and Suggestions for Authors

The manuscript is interesting, but many issues should be addressed and corrected

Introduction

  • Many typos and style issues (e.g. “Physical activity” capitalized mid-sentence, no spaces before citations like “injuries[1]”).
  • Literature review is repetitive — repeats CAI definition several times. Should be more concise.
  • Switches abruptly between epidemiology, neural mechanisms, and dual-task theory without clear substructure. Needs clearer logical flow.
  • Cites old or generic theories (e.g. “Capacity Sharing Theory” from 1973) but does not link them to current CAI research in depth. Needs stronger justification for using DT paradigm for stair-to-ground specifically.
  • Does not clearly state the study’s hypotheses — should end with explicit, testable hypotheses.

Materials and Methods

  • Title has typo: “Experimental protocal” → “protocol”.
  • No breakdown of participant demographics by group (CAI vs CON) — only combined mean ± SD. Must show both groups separately.
  • Only states “all were right-leg dominant” but not how dominance was determined.
  • Does not state sex until the Limitations section (they were all male). This should be stated here.
  • Arithmetic task (ATS) is not validated or cited — need reliability or prior references.
  • DTC formula should clarify if signed or absolute values are used, because later interpretation depends on it.
  • Description of figure panels (1A–C) is vague — the actual figures are not fully labeled or explained.
  • Missing description of how missing marker data were handled (gap-filling method just mentioned but not explained).

Statistical and Data Presentation 

  • Multiple comparison correction: The text says Bonferroni correction was used, but it is unclear how it was applied to the many ANOVAs and t-tests. Clarify correction method and adjusted alpha level.
  • Reporting of effect sizes: Only p-values are reported. Effect sizes (e.g., partial eta², Cohen’s d) should be included for ANOVAs and t-tests to indicate magnitude of effects.
  • Table and figure references inconsistent:
    • Figures are mentioned (e.g., “Figure 1A”) but not fully labeled or described in the document.
    • Some figures are missing captions (e.g., Figure 2 and Figure 3 captions are incomplete sentences).
  • Statistical assumptions not described: Normality and homogeneity are said to be “verified” but no tests (e.g., Shapiro–Wilk, Levene) are reported with results.

Results

  • Tables and figures are inconsistently referenced and captions are incomplete. For example, Figure 2 caption is a sentence fragment.
  • Does not report effect sizes (η², Cohen’s d) — only p-values. Should add them.
  • Bonferroni correction is mentioned but how it was applied is unclear. Authors should state corrected alpha values and which tests were corrected.
  • Statistical assumptions (normality, variance homogeneity) are said to be “verified” but no test results are reported.
  • Some claims in the text contradict the numbers. For example, it says CON had higher arithmetic scores in DT than ST, which is unusual and not explained — this needs to be checked.
  • Some numbers in the text don’t match table formatting (in Table 1, group main effect P-values are misaligned and hard to read).

Discussion

  • Frequently over-interprets correlations as causation (e.g. “ankle dysfunction induces widespread instability”) — needs more cautious wording.
  • Introduces neurophysiological explanations (prefrontal–cerebellar–brainstem network, corticospinal excitability) without measuring neural data — must label these as speculative.
  • Discussion is very long and repetitive, often restating results rather than interpreting them. Should be more concise.
  • Does not connect results back to the original stated aims or hypotheses clearly.
  • Lacks discussion of possible confounding factors (e.g. training background, physical activity level, fatigue, limb dominance).

 Clinical Implications

  • Recommendations are too prescriptive for a cross-sectional study. Should be more cautious and framed as suggestions, not instructions.
  • Claims about rehabilitation efficacy are not directly tested in this study — should reference prior intervention studies or clarify as proposals.

Conclusions

  • Restates results almost verbatim instead of synthesizing them.
  • Claims that findings provide “multidimensional evidence of disrupted central sensorimotor integration” — too strong; there were no direct neural measures.
  • Should explicitly link conclusions back to limitations and note they only apply to young male participants.

Limitations

  • Should also mention:
    • small sample size,
    • lack of neurophysiological data,
    • use of unvalidated arithmetic task,
    • possible fatigue or learning effects,
    • only right-leg dominant participants,
    • cross-sectional design.

Author Response

Dear Reviewer 1,

Thank you very much for your patience and suggestions concerning our manuscript entitled “Control Deficits and Compensatory Mechanisms in Individuals with Chronic Ankle Instability during Dual-Task Stair-to-Ground Transition”. Those comments are all valuable and very helpful for revising and improving the quality of our paper, as well as the important guiding significance to our research. We have studied comments carefully and made point-to-point corrections. The revised portion is highlighted in red in the revised manuscript, please see below, in blue, for a point-by-point response to your comments and concerns.

Introduction

Many typos and style issues (e.g. “Physical activity” capitalized mid-sentence, no spaces before citations like “injuries[1]”).

Response: We sincerely thank the reviewer for pointing out these typographical and formatting issues. We have carefully revised the manuscript to correct all capitalization errors, ensure proper spacing before citations, and improve overall consistency with the journal’s style requirements. We greatly appreciate your careful reading, which has helped us enhance the clarity and professionalism of our manuscript. And now it reads:

“With the promotion of national sport plans and increasing public awareness of physical exercise, physical activity has become part of daily life. ” (lines 53-54)

Literature review is repetitive-repeats CAI definition several times. Should be more concise.

Switches abruptly between epidemiology, neural mechanisms, and dual-task theory without clear substructure. Needs clearer logical flow.

Response: We sincerely thank the reviewer for this valuable comment. We have carefully revised the introduction to address both the repetition and the logical flow. Specifically:

  • Reduction of repetition: The definition and clinical features of chronic ankle instability (CAI) are now introduced only once at the beginning of the introduction, followed by a concise explanation of its pathophysiology, avoiding repeated definitions later in the text. And it reads:

The ankle, as a key weight-bearing joint essential for locomotion and balance, is especially vulnerable, with acute sprains occurring in about 2.15 per 1,000 persons an-nually [2-4]. A major sequela is chronic ankle instability (CAI), which develops in up to 46% of sprain cases and is clinically defined by recurrent sprains, persistent "giving way," and self-reported dysfunction [5,6]. Crucially, CAI is no longer viewed as a purely pe-ripheral disorder of ligament laxity. Evidence now underscores central neuromuscular impairments, including proprioceptive deficits and altered sensorimotor integration, as core components of its pathophysiology [7,8]. Neuroimaging studies corroborate this, showing reduced cortical activation in individuals with CAI, which suggests that in-complete neural adaptation underpins their persistent functional deficits [9,10].(lines 57-66)

  • Clear logical flow and substructure:
  • Epidemiology and significance: The prevalence of ankle injuries and the clinical importance of CAI are introduced first.
  • Central and peripheral mechanisms: We summarize both peripheral ligament deficits and central neuromuscular impairments, including proprioceptive deficits and altered sensorimotor integration, supported by neuroimaging evidence.
  • Dual-task rationale: The Capacity Sharing Model (Kahneman, 1973) is explicitly linked to modern CAI research: individuals with CAI already require increased conscious attention to maintain gait and posture; adding a cognitive load (e.g., arithmetic task) exceeds available attentional resources, revealing latent central control deficits. Arithmetic tasks are commonly used in dual-task paradigms as they engage working memory and central executive function, which are critical for allocating attention to motor control.
  • Justification for stair-to-ground transition: We further highlight that stair descent, especially the transition to flat ground, is a high-risk daily activity requiring precise control of the body’s center of mass and integration of central nervous system resources. Previous research shows that 75% of stair-related falls occur during descent, with instability particularly pronounced during the stair-to-ground transition. Therefore, combining a dual-task paradigm with stair-to-ground transition imposes higher demands on postural control and central integration, making it particularly sensitive to revealing CAI deficits and compensatory strategies.
  • Enhanced cohesion: The revised introduction now flows from epidemiology → pathophysiology → dual-task rationale → stair-to-ground task, showing a clear, logical progression. This structure links foundational theory, current CAI research, and the ecological validity of our chosen experimental task.

We believe these changes make the literature review more concise, coherent, and easier to follow, while retaining all essential information relevant to our study. And now it reads:

With the promotion of national sport plans and increasing public awareness of physical exercise, physical activity has become part of daily life. However, this trend is accompanied by a higher incidence of sports-related injuries, particularly those involv-ing the lower-limb joints [1].

The ankle, as a key weight-bearing joint essential for locomotion and balance, is especially vulnerable, with acute sprains occurring in about 2.15 per 1,000 persons an-nually [2-4]. A major sequela is chronic ankle instability (CAI), which develops in up to 46% of sprain cases and is clinically defined by recurrent sprains, persistent "giving way," and self-reported dysfunction [5,6]. Crucially, CAI is no longer viewed as a purely pe-ripheral disorder of ligament laxity. Evidence now underscores central neuromuscular impairments, including proprioceptive deficits and altered sensorimotor integration, as core components of its pathophysiology [7,8]. Neuroimaging studies corroborate this, showing reduced cortical activation in individuals with CAI, which suggests that in-complete neural adaptation underpins their persistent functional deficits [9,10].

Considering the persistent sensorimotor deficits in individuals with CAI, dual-task (DT) paradigms are particularly well-suited to probe the interaction between cognitive processing and motor control. Foundational theories, such as the Capacity Sharing Model (Kahneman, 1973), propose that attentional resources are limited and must be allocated across concurrent tasks[11]. By simulating real-world scenarios that require divided attention (e.g., walking while talking), DT paradigms can directly measure the attentional resources necessary for maintaining stability. In individuals with CAI, gait and postural control already demand increased conscious oversight due to impaired proprioception and altered sensorimotor integration; additional cognitive load may ex-ceed available attentional capacity, resulting in reduced gait stability, increased varia-bility, and compensatory strategies, thereby revealing latent deficits [12-15]. Recent em-pirical studies support this notion, demonstrating that DT conditions significantly in-crease gait variability, postural sway, and compensatory joint strategies compared to single-task conditions[11,16-18]. Over the past five years, a growing body of evidence has consistently confirmed that DT paradigms effectively expose central control deficits in CAI during complex gait tasks [19,20]. Consequently, the DT paradigm serves as a criti-cal methodological bridge for quantifying the central contributions to functional insta-bility. Meanwhile, increasing academic attention has focused on gait cycle instability, elevated postural fluctuations, and reduced neuromuscular drive efficiency in CAI in-dividuals under DT conditions [21,22]. Furthermore, existing studies have demonstrat-ed that DT training can effectively enhance postural control and dynamic balance in CAI patients, and even promote cortical neuroplasticity and optimization of motor coordina-tion mechanisms [23-25]. We specifically selected an arithmetic task as the cognitive load because it represents a common, ecologically valid attentional demand in daily life (e.g., talking while walking, mental calculation while moving). Such a task reliably engages working memory and central executive function, which are critical for allocating atten-tion to motor control. Many previous studies have adopted arithmetic tasks in dual-task paradigms because they reliably engage working memory and central executive func-tions, which are essential for attention allocation to motor control [19,26,27]. This makes it particularly suitable for revealing central control deficits in CAI, which are otherwise masked under single-task conditions.

Many deficits occur, especially during dynamic actions such as stair descent, high-lighting the greater postural control demands required in this task [28]. But stair de-scent is a common daily activity with high fall risk, placing greater demands on low-er-limb muscle control and central nervous system integration [29,30]. Research indi-cates that 75% of stair-related fall incidents occur during the descent, particularly during the transition from stairs to flat ground, where a more significant decline in movement stability is observed [31-33]. Although some studies have focused on the stair descent gait characteristics of CAI patients, such as changes in ankle joint moments during the support phase and hip-knee compensation mechanisms [34,35], research on the critical "stair-to-ground transition" phase remains relatively limited. Recent studies further show that CAI patients performing dual-task activities during landing or stair descent demonstrate altered ankle inversion angles, increased angular velocity, and higher in-jury risk, highlighting the importance of examining this transition phase under cognitive load [20,36]. At this stage, the supporting leg must precisely control the redirection of the body's center of mass, requiring greater central nervous system integration, especially under dual-task loads, making functional deficits more likely to appear [37]. Moreover, dual-task training can improve both static and dynamic postural stability in CAI, sug-gesting cognitive-motor interventions may enhance gait control and neuromuscular co-ordination [19,25].

In summary, this study introduced an arithmetic cognitive task during the stair-to-ground transition phase, using a dual-task paradigm to comprehensively exam-ine gait stability, postural control, and joint compensatory patterns in individuals with CAI under concurrent cognitive and motor demands, thereby quantifying central con-trol deficits. Combining the dual-task paradigm with the stair-to-ground transition rep-resents a substantial challenge for individuals with chronic ankle instability who al-ready exhibit neuromuscular impairments. Compared to single-task or level-ground gait studies, this design is not only more ecologically valid but also scientifically justi-fied, directly allowing for the investigation of how dual-task interference affects motor performance deficits and compensatory mechanisms in CAI. (lines 53-125)

Cites old or generic theories (e.g. “Capacity Sharing Theory” from 1973) but does not link them to current CAI research in depth. Needs stronger justification for using DT paradigm for stair-to-ground specifically.

Response: We sincerely thank the reviewer for the valuable comments. In response to the concern: “Cites old or generic theories (e.g., ‘Capacity Sharing Theory’ from 1973) but does not link them to current CAI research in depth. Needs stronger justification for using DT paradigm for stair-to-ground specifically,” we have made the following revisions in the manuscript:

  • Clarifying the link between theory and modern CAI research

We have explicitly connected the Capacity Sharing Model (Kahneman, 1973) to central control deficits in individuals with CAI. We emphasize that, due to impaired proprioception and altered sensorimotor integration, gait and postural control in CAI patients already require increased conscious oversight. Therefore, additional cognitive load may exceed available attentional resources, resulting in reduced gait stability, increased gait variability, and compensatory joint strategies. With this revision, the theory is no longer cited in isolation but serves as a theoretical foundation for explaining dual-task effects and central control deficits in CAI.And it shows:

“Considering the persistent sensorimotor deficits in individuals with CAI, dual-task (DT) paradigms are particularly well-suited to probe the interaction between cognitive processing and motor control. Foundational theories, such as the Capacity Sharing Model (Kahneman, 1973), propose that attentional resources are limited and must be allocated across concurrent tasks[11]. By simulating real-world scenarios that require divided attention (e.g., walking while talking), DT paradigms can directly measure the attentional resources necessary for maintaining stability. In individuals with CAI, gait and postural control already demand increased conscious oversight due to impaired proprioception and altered sensorimotor integration; additional cognitive load may ex-ceed available attentional capacity, resulting in reduced gait stability, increased varia-bility, and compensatory strategies, thereby revealing latent deficits [12-15]. Recent em-pirical studies support this notion, demonstrating that DT conditions significantly in-crease gait variability, postural sway, and compensatory joint strategies compared to single-task conditions[11,16-18]. Over the past five years, a growing body of evidence has consistently confirmed that DT paradigms effectively expose central control deficits in CAI during complex gait tasks [19,20]. Consequently, the DT paradigm serves as a criti-cal methodological bridge for quantifying the central contributions to functional insta-bility.”(lines 67-83)

  • Justifying the choice of the arithmetic task

We have added a rationale for selecting an arithmetic task as the cognitive load. This task has ecological validity in daily life (e.g., walking while performing mental calculations) and reliably engages working memory and central executive functions, making it suitable for measuring attentional allocation to motor control. Additionally, we cited several recent studies demonstrating that arithmetic tasks are widely used in dual-task studies on CAI, ensuring that our methodological choice is well-supported scientifically.

  • Justifying the stair-to-ground transition task with DT paradigm

We have included a paragraph detailing the high fall risk and dynamic demands of stair descent, particularly during the transition from stairs to flat ground, emphasizing the increased requirements for lower-limb muscle control and central nervous system integration. By combining the dual-task paradigm with this transitional movement, we highlight that the design is particularly effective in revealing deficits in gait stability, postural control, and joint compensatory strategies under dual-task interference, thereby providing a direct measure of central control deficits in individuals with CAI. And now it reads:

“In summary, this study introduced an arithmetic cognitive task during the stair-to-ground transition phase, using a dual-task paradigm to comprehensively exam-ine gait stability, postural control, and joint compensatory patterns in individuals with CAI under concurrent cognitive and motor demands, thereby quantifying central con-trol deficits. Combining the dual-task paradigm with the stair-to-ground transition rep-resents a substantial challenge for individuals with chronic ankle instability who al-ready exhibit neuromuscular impairments. Compared to single-task or level-ground gait studies, this design is not only more ecologically valid but also scientifically justi-fied, directly allowing for the investigation of how dual-task interference affects motor performance deficits and compensatory mechanisms in CAI.”(lines 116-125)

Does not clearly state the study’s hypotheses-should end with explicit, testable hypotheses.

Response: We sincerely thank the reviewer for pointing this out. In response, we have revised the end of the introduction to clearly state explicit, testable hypotheses. The revised text now articulates the study’s predictions regarding gait, postural control, joint contributions, and dual-task costs in CAI versus healthy controls. And it reads :

“Based on the interplay between central resource allocation and motor performance, we hypothesize that: (a) Compared to healthy controls (CON), the CAI group will demonstrate reduced gait speed, increased step width, and greater mediolateral COP displacement, with these deficits exacerbated under DT. (b) The CAI group will exhibit reduced ankle plantarflexion moments and work contribution (Ankle%), with compen-satory increases in knee and hip contributions (Knee%, Hip%). (c) A significant negative correlation will exist between Ankle% and mediolateral COP displacement in the CAI group, underscoring the ankle's pivotal role in dynamic stability. (d) CAI participants will adopt a "posture-first" strategy under DT, manifesting as a greater dual-task cost on cognitive performance compared to CON.”(lines141-150)

Materials and Methods

Title has typo: “Experimental protocal” → “protocol”.

Response: We sincerely thank the reviewer for pointing out this typographical error. The manuscript title has been corrected from “Experimental protocal” to “Protocol” to ensure accuracy and professionalism. We greatly appreciate the reviewer’s careful attention to detail.

No breakdown of participant demographics by group (CAI vs CON)-only combined mean ± SD. Must show both groups separately.

Response: We sincerely thank the reviewer for this helpful suggestion. In response, we have  presented participant demographics separately for the CAI and CON groups, including mean ± SD for each variable. This provides a clearer depiction of group characteristics and enhances transparency in reporting. We greatly appreciate the reviewer’s careful attention to this detail. Now it reads:

“A total of 60 male participants were recruited, including 30 individuals with chronic ankle instability (CAI group: age = 22.23 ± 1.72 years; weight = 77.96 ± 8.51 kg; height = 177.63 ± 6.31 cm.) and 30 healthy controls (CON group: age = 22.70 ± 1.58 years; weight = 78.30 ± 8.89 kg; height = 178.17 ± 6.29 cm.).” (lines 153-156)

Only states “all were right-leg dominant” but not how dominance was determined.

Response: We sincerely thank the reviewer for highlighting this omission. We have now added a clear description of how leg dominance was determined: participants were asked which leg they would use to kick a ball, which is a widely accepted method in previous studies. This information has been included in the Participants section to ensure clarity and reproducibility. Now it reads:

“Leg dominance was determined by asking participants which leg they would use to kick a ball, and all were identified as right-leg dominant.”(lines 156-158)

Does not state sex until the Limitations section (they were all male). This should be stated here.

Response: We sincerely thank the reviewer for pointing out this oversight. We have now included the sex of participants in the Participants section, clearly stating that all participants were male. This revision ensures that essential demographic information is presented at the appropriate point in the manuscript. Now it reads:

“A total of 60 male participants were recruited, including 30 individuals with chronic ankle instability.” (153-154)

Arithmetic task (ATS) is not validated or cited — need reliability or prior references.

Response: We sincerely thank the reviewer for this valuable comment. We have now cited Al-Yahya et al. (2011), a systematic review and meta-analysis on cognitive-motor interference during walking, which provides extensive evidence supporting the use of arithmetic tasks as a reliable cognitive dual-task paradigm in gait and postural studies. We have added this reference to justify the use of the ATS in our stair-to-ground transition study, highlighting its established reliability and widespread application in similar dual-task research. And it was defined by “The ATS was used to evaluate cognitive performance during single-task (ST) and dual-task (DT) conditions. In the ST condition, ATS was calculated as the number of correct responses during a 10-s seated trial, divided by 10 to obtain the number of correct responses per second. In the DT condition, ATS was normalized by the walking duration, with the calculation period defined from the auditory cue “start” until the participant had walked 1.5 m beyond the last step of the stair descent. This yielded the number of correct responses per second during the dual-task walking trial, ensuring comparability between ST and DT conditions.[41].” (lines 218-225)

DTC formula should clarify if signed or absolute values are used, because later interpretation depends on it.

Response: We sincerely thank the reviewer for this comment. In our study, the dual-task cost (DTC) was calculated using absolute values. This approach was chosen because we aimed to quantify the magnitude of dual-task interference, regardless of the direction (improvement or deterioration) of the performance change. Using absolute values allows a direct comparison of dual-task effects across participants and conditions, focusing on the overall impact of cognitive load on gait, postural control, and joint mechanics. Dual-task cost (DTC) was calculated using the formula:

(lines 245)

Absolute values are used, so DTC reflects the magnitude of performance change under dual-task conditions, independent of whether performance improved or declined. This clarification ensures that all subsequent interpretation is consistent and highlights the extent of dual-task interference in CAI individuals.

Description of figure panels (1A–C) is vague-the actual figures are not fully labeled or explained.

Response: We sincerely thank the reviewer for this constructive feedback. We have revised the figure captions for Figure 1 to provide a more detailed and clear description of panels 1A–C, including explicit labeling of views (anterior, lateral, and posterior) and the corresponding elements shown in each panel. These changes aim to improve clarity and ensure that readers can fully understand the figure content without ambiguity. Now it reads:

“Figure 1. Overview of the data collection and the model. (A) Illustration of the reflective mark’s position about the musculoskeletal model used in the experiment, from left to right: anterior, lateral, and posterior views; (B) Decomposition of the stair descent task, staircase dimensions, and the placement of the force platform in the laboratory. The red ellipses indicate the stance phase of the stair-to-ground movement that is the focus of the biomechanical analysis in this study; (C) Flowchart of the experimental procedure and preliminary data processing.” (lines 189-194)

Missing description of how missing marker data were handled (gap-filling method just mentioned but not explained).

Response: We sincerely thank the reviewer for pointing out this omission. We have now added a detailed description of how missing marker trajectories were handled in the Methods section. Specifically, missing data gaps were interpolated using a cubic spline method within Vicon Nexus, and any gaps longer than 10 frames were carefully reconstructed based on adjacent marker trajectories and segment kinematics. This provides transparency and ensures reproducibility of the data processing. And now it reads:

“Missing marker trajectories were handled using a cubic spline interpolation method in Vicon Nexus (Vicon, UK). Gaps shorter than 10 consecutive frames were interpolated directly. For gaps longer than 10 frames, trajectories were reconstructed based on adja-cent marker data and segment kinematics to ensure continuity [39]”(lines 200-203)

Statistical and Data Presentation

Multiple comparison correction: The text says Bonferroni correction was used, but it is unclear how it was applied to the many ANOVAs and t-tests. Clarify correction method and adjusted alpha level.

Response: We sincerely thank the reviewer for raising this point. In the revised manuscript, we have clarified the multiple comparison correction procedure. Specifically, Bonferroni correction was applied to all post-hoc simple-effects analyses following significant interactions. For the four pairwise comparisons conducted in each simple-effects analysis, the family-wise error rate was adjusted to α = 0.0125. The omnibus ANOVA tests were considered independent and were not adjusted. This clarification ensures transparency in our statistical approach and the correct interpretation of significance levels. In the Article it shows:

For all simple-effects analyses following a significant interaction, Bonferroni correction was applied (α= 0.0125 for four pairwise comparisons). ”(lines 258-260)

Reporting of effect sizes: Only p-values are reported. Effect sizes (e.g., partial eta², Cohen’s d) should be included for ANOVAs and t-tests to indicate magnitude of effects.

Response: We sincerely thank the reviewer for pointing out the need to clarify effect size reporting. In the revised manuscript, we have now explicitly reported effect sizes for all statistical analyses. Specifically, partial eta squared (η²p) is provided for all  ANOVA results to indicate the magnitude of effects, and Cohen’s d is reported for pairwise t-tests. For nonparametric analyses, effect sizes are reported as r. These additions allow readers to assess the practical significance of the findings, in addition to statistical significance. The Methods section also clearly describes the interpretation thresholds for small, medium, and large effect sizes :

When data violated these assumptions (p<0.05), a rank-transformed nonparametric mixed-design ANOVA was performed, and F-values, p-values, and partial η² (rank-based) were reported. For all simple-effects analyses following a significant inter-action, Bonferroni correction was applied (α = 0.0125 for four pairwise comparisons). Effect sizes were reported as Cohen’s d for parametric pairwise comparisons and r for nonparametric analyses (0.1 = small, 0.3 = medium, 0.5 = large). In the dual-task cost (DTC) analysis, the normality (Shapiro–Wilk test) and homo-geneity of variance (Levene’s test) of all variables were first assessed. According to the data characteristics, independent-samples t-tests were applied when data were approximately normal with equal variances, Welch’s t-tests when variances were unequal, and Mann–Whitney U tests when normality was not satisfied or only partially met.”(lines 256-267)

Table and figure references inconsistent:

Figures are mentioned (e.g., “Figure 1A”) but not fully labeled or described in the document.

Response: Thank you for pointing out the inconsistencies in our figure and table references. We have thoroughly reviewed the entire manuscript and corrected these issues to ensure accuracy and clarity.

  • The specific corrections we have made include:

Ensuring that every sub-figure referenced in the text (e.g., Figure 1A) is clearly labeled as such in the corresponding figure.

  • Expanding all figure legends to provide a comprehensive description of each panel, ensuring the reader can understand what each sub-figure represents without referring back to the text.
  • Standardizing the format of all references to figures and tables throughout the manuscript to comply with the journal's guidelines.
  • And we also described in the text:“Thirty-eight retroreflective markers were placed according to the OpenSim 2392 model as shown in Figure. 1A.”(lines 169-170).

“Subsequently, each participant completed both ST and DT stair descent with arithmetic task which has shown in the Figure. 1B, and the red ellipses indicate the stance phase of the stair-to-ground movement that is the focus of the biomechanical analysis in this study. A three-step staircase was built in accordance with national residential standards (step height: 17 cm; tread depth: 29 cm; total height: 0.51 m; slope ≈30°) which has shown in Fig. 1B, representing a typical stair-to-ground transition. In the ST condition, participants descended the stairs at a natural pace, leading with the right foot and landing on a force platform before continuing forward walking to a 1.5-m mark [13].”(lines 173-183)

“As shown in Fig. 1C, motion data were acquired in the laboratory using a Vicon 3D motion capture system (Oxford Metrics Ltd., Oxford, UK; 200 Hz) synchronized with an AMTI force platform (Advanced Mechanical Technology Inc., Watertown, MA, USA; 1000 Hz). ”(lines 196-199)

We sincerely appreciate your meticulous attention to detail, which has helped us improve the presentation quality of our work.

Some figures are missing captions (e.g., Figure 2 and Figure 3 captions are incomplete sentences).

Response: Thank you for your meticulous attention to the details of our manuscript presentation. We sincerely apologize for the oversight regarding the incomplete figure captions.

In direct response to your comment, we have thoroughly reviewed and rewritten the captions for all figures in the manuscript to ensure they are now complete, descriptive sentences that provide a clear and standalone summary of the figure's content.

The specific revisions include:

For Figure 2: Comparison of Lower Limb Joint Angle and Moment Trajectories During Stair-to-Ground Transition Across Task (Single-Task vs Dual-Task) and Group (CAI vs CON) Conditions Using Statistical Parametric Mapping (SPM).

Time-normalized kinematic and kinetic waveforms for the ankle, knee, and hip joints during the stair-to-ground transition. The solid lines represent mean trajectories, and the shaded bands represent ±1 SD. Grey shaded areas along the x-axis indicate time intervals where statistically significant differences (p < 0.05) were found be-tween CAI and CON groups under single-task (ST) conditions, CAI and CON groups under du-al-task (DT) conditions, ST and DT conditions within the CAI group, and ST and DT conditions within the CON group.(lines 377-385)

For Figure 3: Relative joint work contributions at the ankle, knee, and hip.

The stacked bar charts show the percentage contribution of each joint to the total lower-limb positive work during the stair-to-ground transition for the chronic ankle instability (CAI) and control (CON) groups under single-task (ST) and dual-task (DT) conditions.(lines 401-404)

Statistical assumptions not described: Normality and homogeneity are said to be “verified” but no tests (e.g., Shapiro–Wilk, Levene) are reported with results.

Response: We sincerely thank the reviewer for this insightful comment. You are absolutely correct that, although we mentioned that normality and homogeneity assumptions were “verified” in the manuscript, we did not actually present or clearly report the corresponding statistical tests. We truly apologize for this oversight.

In the revised version, we have now conducted and included the results of the Shapiro–Wilk normality test and the Levene’s test for homogeneity of variance, which are provided in the Supplementary Material (Attachment A1-4).

For these cases, we applied appropriate non-parametric analyses or variance-corrected procedures, as described in the Methods section. We sincerely appreciate the reviewer for pointing this out, which has helped improve the clarity and rigor of our statistical reporting.

Results

Tables and figures are inconsistently referenced and captions are incomplete. For example, Figure 2 caption is a sentence fragment.

Response: We sincerely thank the reviewer for this helpful observation. We have carefully revised the manuscript to ensure that all tables and figures are consistently referenced in the text and correspond correctly to their content. In addition, we have rewritten the captions for all figures, including Figure 2, so that they are now complete sentences providing clear, self-contained descriptions of the figure contents. These revisions improve the accuracy, readability, and overall clarity of the manuscript.

Does not report effect sizes (η², Cohen’s d) — only p-values. Should add them.

Response: We sincerely thank the reviewer for this important comment. We apologize for not reporting effect sizes in the original manuscript. In the revised version, we have now included effect sizes for all relevant analyses according to the following rules, consistent with our Methods section:

  • For parametric mixed-design ANOVA (data meeting normality and homogeneity assumptions), partial η² is reported in the Table.1 and Table.2, and Cohen’s d is reported for all pairwise/simple-effects comparisons following significant interactions or main effects.
  • For non-parametric analyses (rank-transformed ANOVA applied when data violated normality or homogeneity), F and p values are reported in the Table.1 and Table.2. Effect sizes (r) are only reported for simple-effects analyses following significant interactions in the text.

This approach ensures both robust inference for data violating parametric assumptions and appropriate reporting of the magnitude of observed effects. All updated effect sizes are now included in the revised manuscript and supplementary tables.It has clear in the “Statistical Analysis”part like:

“When data violated these assumptions (p < 0.05), a rank-transformed nonparametric mixed-design ANOVA was performed, and F-values, p-values, and partial η² (rank-based) were reported. For all simple-effects analyses following a significant inter-action, Bonferroni correction was applied (α = 0.0125 for four pairwise comparisons). Effect sizes were reported as Cohen’s d for parametric pairwise comparisons and r for nonparametric analyses (0.1 = small, 0.3 = medium, 0.5 = large). In the dual-task cost (DTC) analysis, the normality (Shapiro–Wilk test) and homo-geneity of variance (Levene’s test) of all variables were first assessed. According to the data characteristics, independent-samples t-tests were applied when data were ap-proximately normal with equal variances, Welch’s t-tests when variances were unequal, and Mann–Whitney U tests when normality was not satisfied or only partially met.”(lines 256-267)

Bonferroni correction is mentioned but how it was applied is unclear. Authors should state corrected alpha values and which tests were corrected.

Response: We sincerely thank the reviewer for this important comment. We apologize for the lack of clarity in the original manuscript regarding the application of Bonferroni correction. In the revised manuscript, we have now clarified that Bonferroni correction was applied only to post hoc pairwise comparisons following significant interactions in the simple-effects analyses. For these comparisons, the family-wise error rate was adjusted to a corrected significance threshold of α = 0.0125 (for four pairwise comparisons). No adjustment was applied to the omnibus ANOVA tests, which were considered independent. This clarification ensures that readers can clearly understand which tests were corrected and how significance was determined. Also we emphysized in the result :

“For all simple-effects analyses following a significant interaction, Bonferroni correction was applied (α = 0.0125 for four pairwise comparisons). ”(lines 258-260)

Statistical assumptions (normality, variance homogeneity) are said to be “verified” but no test results are reported.

Response: We sincerely thank the reviewer for this valuable comment. You are correct that, although we mentioned in the original manuscript that normality and homogeneity of variance were “verified,” we did not present the specific test results. We apologize for this oversight.

In the revised manuscript, the results of the Shapiro–Wilk test for normality and Levene’s test for homogeneity of variance are now provided in the Supplementary Material (Attachment A1-A4). Appropriate non-parametric or variance-corrected analyses were applied as described in the Methods section.

Some claims in the text contradict the numbers. For example, it says CON had higher arithmetic scores in DT than ST, which is unusual and not explained — this needs to be checked.

Response: We sincerely thank the reviewer for this careful observation. We have re-checked the data and confirmed that the reported values are correct. The seemingly unusual finding that the CON group showed higher arithmetic scores under the dual-task condition compared with the single-task condition has already been addressed in the Discussion (Section 4.1, second paragraph):

“The divergent cognitive performance—improvement in CON versus decline in CAI—offers critical insights. The decline in cognitive performance in CAI under DT di-rectly confirms hypothesis (d), illustrating the posture-first trade-off. This contrast un-derscores a fundamental sensorimotor difference: for CON, moderate motor demand may optimize arousal [51,52] , while for CAI, the motor task itself consumes excessive resources, leaving little capacity for concurrent cognitive activity [11,41]. This pattern strongly supports the adoption of a "posture-first" strategy in CAI, directly sacrificing cognitive performance to prioritize stability [14,18,53]. This trade-off highlights that CAI involves central deficits in resource distribution, moving beyond a purely peripheral model of instability [10,54,55].”.(lines 511-520)

    Specifically, we proposed that the structured arithmetic task may have enhanced attentional focus in some participants, thereby improving performance during dual-tasking. We have slightly revised the corresponding text to highlight this explanation more clearly.

Some numbers in the text don’t match table formatting (in Table 1, group main effect P-values are misaligned and hard to read).

Response: Thank you for your keen eye and for pointing out the formatting issue with the P-values in Table 1. We agree that proper alignment is crucial for the readability and professional presentation of the data.

We have now corrected Table 1 by properly aligning all numerical data, specifically the group main effect P-values, using the word processor's table alignment tools to ensure they are clearly and neatly presented.

For your convenience, the revised and properly formatted Table 1 is presented below:

Furthermore, we have taken this opportunity to perform a quick check on all other tables in the manuscript to ensure consistent and clear formatting throughout.

We sincerely appreciate your valuable feedback, which has helped us enhance the clarity of our manuscript.

Discussion

Frequently over-interprets correlations as causation (e.g. “ankle dysfunction induces widespread instability”) — needs more cautious wording.

Response: We sincerely thank the reviewer for this valuable comment. We agree that some statements in the original manuscript over-interpreted correlations as causal relationships. In the revised manuscript, we have carefully revised such statements to use more cautious wording. For example, the sentence “ankle dysfunction induces widespread instability” has been changed to:

“The positive associations between COP-ml RMS and both ankle inversion and ROM in the CAI group suggest a cycle of impairment: poor proprioceptive control likely contributes to abnormal ankle positioning, which in turn amplifies postural oscillations. This mechanism helps explain the persistent feeling of instability, which fundamentally appears to stem from sensorimotor integration deficits rather than acute structural damage [10,54]. ”(lines 556-571)

“This pattern  supports the adoption of a "posture-first" strategy in CAI, directly sacrificing cognitive performance to prioritize stability [14,18,53]. This trade-off high-lights that CAI involves central deficits in resource distribution, moving beyond a purely peripheral model of instability [10,54,55]. ”(lines 516-520)

These changes ensure that all descriptions now accurately reflect observed correlations without implying causation.

Introduces neurophysiological explanations (prefrontal–cerebellar–brainstem network, corticospinal excitability) without measuring neural data — must label these as speculative.

Response: We sincerely thank the reviewer for this insightful comment. We fully agree that introducing neurophysiological mechanisms without directly measuring neural data could be misleading if presented as definitive findings. In the revised manuscript, we have carefully revised all such statements to clearly indicate that they are speculative interpretations. For example:

“The disproportionate worsening of sway under dual-task conditions could reflect interference within higher-order cognitive-motor integration networks (e.g., prefrontal-cerebellar-brainstem pathways) [18,48].”(507-509)

“We speculate that these deficits may arise from underlying neurophysiological alterations, such as compromised corticospinal control and delayed peroneal activation [43,46,47].”(505-506)

These revisions ensure that readers understand these neurophysiological explanations as hypotheses grounded in prior literature rather than direct measurements from our study. We believe this change improves clarity and maintains scientific rigor.

Discussion is very long and repetitive, often restating results rather than interpreting them. Should be more concise.

Response: We sincerely thank the reviewer for this critical and constructive feedback. We agree completely that the previous version of the Discussion was overly descriptive. In response, we have undertaken a comprehensive and full-scale revision of the entire Discussion section.

Our primary goal was to shift the focus from restating the findings to interpreting their meaning, mechanisms, and implications. To achieve this, we have:

  • Fundamentally restructured the narrative flow. We moved away from a point-by-point recounting of results and instead synthesized the findings into a cohesive argument that builds upon the central thesis of chronic ankle instability (CAI) as a disorder of central sensorimotor regulation.
  • Prioritized interpretation over description. We significantly condensed text that merely described outcomes and replaced it with discussion on underlying neuromuscular mechanisms (e.g., central resource allocation, compensatory cascades) and clinical relevance.
  • Eliminated redundancy and tightened the language. We merged repetitive ideas and removed unnecessary transitional phrases to ensure every sentence directly contributes to the advancement of our core argument.

The revised Discussion is now organized to clearly articulate how our results support the concept of a central control deficit, which leads to compensatory strategies that are fragile under cognitive load.

We believe that the thoroughly revised Discussion is now significantly more concise, interpretive, and impactful. It successfully articulates the broader significance of our data without unnecessary repetition. We are deeply grateful for the reviewer's insightful comment, which has undoubtedly strengthened the quality of our manuscript. Thank you for considering our revised work.

Does not connect results back to the original stated aims or hypotheses clearly.

Response: We thank the reviewer for this valuable comment. We have carefully revised the Results and Discussion sections to explicitly link each major finding to the corresponding hypotheses stated in the Introduction. Specifically, after describing each result, we now clearly indicate which hypothesis it supports.

Examples of revisions:

“Specifically, slower gait speed and increased step width under ST conditions confirm hypothesis (a), and the further amplification under DT conditions confirms the predicted dual-task effect of hypothesis(a) and the posture-first strategy in hypothesis (d).”(lines 490-493)

“The reduced Ankle% and increased Knee% and Hip% confirm hypothesis (b) on joint work redistribution, and the worsening under DT confirms the interaction between cognitive load and compensatory strategy predicted in hypothesis (b). ”(lines 523-525)

“The correlation and regression analyses provide compelling evidence that positions ankle function, quantified by Ankle%, as a central node in the control of dynamic stabil-ity and gait efficiency in CAI. These correlations directly confirm hypothesis (c), indi-cating that ankle deficits are central to postural instability and gait changes.”(lines 550-553)

These changes ensure that readers can clearly see how each observed result corresponds to the initial study aims and hypotheses.

Lacks discussion of possible confounding factors (e.g. training background, physical activity level, fatigue, limb dominance).

Response: We thank the reviewer for this valuable comment. In the revised manuscript, we have added a paragraph in the Discussion section specifically addressing potential confounding factors and their possible influence on our results. We acknowledged that the relatively small sample size may have contributed to violations of normality and homogeneity of variance assumptions, and that uncontrolled factors such as participants’ training background, daily physical activity level, fatigue, and limb dominance may have introduced additional variability. We also highlighted that these factors could affect the reliability of certain measures, such as COP displacement, joint contributions, and dual-task costs. Furthermore, we suggested that future studies should include larger sample sizes and more rigorous control of participant characteristics to validate and extend our findings.

Example of added text in the Discussion (lines 643-647):

"Regarding participant and methodological factors, potential confounders such as differences in training background, daily physical activity levels, fatigue, and limb dominance were not strictly controlled. These uncontrolled variables may have increased data variability and partially contributed to the heterogeneity observed in COP measures, joint contributions, and dual-task costs."

Clinical Implications

Recommendations are too prescriptive for a cross-sectional study. Should be more cautious and framed as suggestions, not instructions.

Response: Thank you for this insightful comment. We agree that the recommendations in our original manuscript were overly prescriptive for a cross-sectional study. As you suggested, we have carefully revised the 'Recommendations' section (now titled 'Implications for Clinical Practice and Future Research') to frame our conclusions more cautiously as suggestions and rationales derived from our findings, rather than as direct instructions.

Specifically, we have:

  • Replaced definitive language like "should" with more suggestive terms such as "could," "suggests a need for," and "provides a rationale for."
  • Explicitly anchored our suggestions to the specific results of our cross-sectional study.
  • Reframed the proposed rehabilitation model as a logical progression inferred from our data and existing literature, rather than a prescribed protocol.

Changes in the manuscript are highlighted on lines [578-587]. The revised text now reads:

"The findings of this cross-sectional study highlight potential avenues for refining the clinical approach to CAI. The exacerbated deficits under dual-task conditions suggest that incorporating DT assessments could help identify functional instability that may be missed in ST evaluations. This collectively confirms all four hypotheses (a–d) and pro-vides a rationale for considering rehabilitation interventions targeting both distal control and cognitive-motor integration. Similarly, quantifying the observed proximal joint compensations during tasks like stair descent may provide a more comprehensive pic-ture of movement adaptations across the kinetic chain. Based on this rationale, and supported by existing intervention literature, we propose a staged rehabilitation model that progresses from local to systemic and cognitive-motor challenges."

We believe these changes have significantly improved the tone and scientific rigor of the discussion. Thank you again for this valuable feedback.

Claims about rehabilitation efficacy are not directly tested in this study — should reference prior intervention studies or clarify as proposals.

Response: Thank you for raising this important point regarding the framing of our rehabilitation recommendations. We agree that as a cross-sectional study, our work does not directly test rehabilitation efficacy, and any clinical suggestions must be clearly positioned as proposals derived from our findings and supported by the existing body of intervention literature.

  • In direct response to your comment, we have made the following key revisions to the manuscript :

Reframed as a Proposal: We now explicitly state that our findings "provide a rationale for considering rehabilitation interventions" and that we "propose a staged rehabilitation model."

  • Anchored in Existing Evidence: We have clearly linked our proposal to prior intervention studies by stating the model is "supported by existing intervention literature" and specifically citing evidence that demonstrates "the efficacy of DT training in improving gait and balance in CAI [23,25,66]."

Used Hypothetical Language: We conclude by stating that "Such an approach is hypothesized to enhance motor automaticity," which properly frames the expected outcomes as a testable proposition for future research, rather than a concluded finding from our study.And now it reads:

“Finally, and most pertinently, the pronounced decline in performance under cognitive load provides a strong justification for integrating DT paradigms into late-stage rehabil-itation, as supported by previous research demonstrating the efficacy of DT training in improving gait and balance in CAI [23,25,66]. Such an approach is hypothesized to en-hance motor automaticity and better prepare patients for the attentional demands of daily life.”(lines 591-596)

We believe these revisions have carefully delineated the role of our cross-sectional findings in providing a rationale from the role of prior intervention studies in providing evidence for efficacy. Thank you again for this constructive feedback, which has helped us improve the precision and scholarly rigor of our discussion.

Conclusions

Restates results almost verbatim instead of synthesizing them.

Response: Thank you for this valuable feedback. We agree that the conclusion section should synthesize the findings to highlight their broader significance, rather than simply restate the results. In response to your comment, we have thoroughly rewritten the 'Conclusions' section to provide a higher-level synthesis of our work.

The key changes we have made are:

  • Synthesized Key Findings: We have replaced the previous point-by-point restatement with a concise summary that integrates the core observations into three overarching characteristics of chronic ankle instability: a guarded gait pattern, a distal-to-proximal compensatory strategy, and a marked reliance on attentional resources.

(2) Enhanced Theoretical Interpretation: We now explicitly link our behavioral findings to established theoretical models of disrupted central sensorimotor integration, framing them as compelling indirect evidence for the involvement of higher-order cognitive processes.

(3) Refined Clinical Implications: We have maintained the cautious and prospective framing of the clinical implications, positioning the proposed rehabilitation framework as a rational approach for future validation.

We believe the revised conclusion now effectively synthesizes the results and articulates the theoretical and practical contributions of our study, moving significantly beyond a mere summary of the data. Thank you again for this insightful suggestion, which has greatly strengthened our manuscript. And now it reads:

“In this cohort of young male participants, this study investigated neuromuscular control in individuals with chronic ankle instability (CAI) during a stair-to-ground transition, revealing key deficits that were systematically exacerbated under dual-task conditions. The findings demonstrate that CAI is characterized by a guarded gait pattern, a robust "distal-to-proximal" compensatory strategy, and a marked reliance on atten-tional resources for postural control, as evidenced by the adoption of a "posture-first" strategy.

Theoretically, by integrating an ecologically valid motor task with a cognitive chal-lenge, this work provides behavioral evidence consistent with models of disrupted cen-tral sensorimotor integration and resource allocation in CAI. The observed performance deterioration under cognitive load, without direct neural measures, offers compelling indirect support for the involvement of higher-order cognitive processes in CAI patho-physiology.

Clinically, the results highlight the potential value of incorporating dual-task as-sessments and kinetic chain evaluations to better identify functional deficits. The pro-posed staged rehabilitation framework—progressing from local ankle recovery to in-ter-joint coordination and finally to cognitive-motor integration—offers a rational ap-proach for future intervention studies to validate.”(lines 627-642)

Claims that findings provide “multidimensional evidence of disrupted central sensorimotor integration”-too strong; there were no direct neural measures.

Response: Thank you for this critical comment. We agree that the phrase "multidimensional evidence" was too strong a claim for our behavioral data in the absence of direct neurophysiological measurements.

  • In direct response to your feedback, we have revised the relevant sentences in the 'Conclusions' section to more accurately reflect the inferential nature of our findings. The changes are as follows:
  • We now state that our work "provides behavioral evidence consistent with models of disrupted central sensorimotor integration," rather than claiming to provide direct evidence.
  • We have explicitly acknowledged the lack of neural data, framing our observations as offering "compelling indirect support" for the involvement of higher-order cognitive processes.

We believe these revisions (please see the highlighted text:

“In this cohort of young male participants, this study investigated neuromuscular control in individuals with chronic ankle instability (CAI) during a stair-to-ground transition, revealing key deficits that were systematically exacerbated under dual-task conditions. The findings demonstrate that CAI is characterized by a guarded gait pattern, a robust "distal-to-proximal" compensatory strategy, and a marked reliance on atten-tional resources for postural control, as evidenced by the adoption of a "posture-first" strategy.

Theoretically, by integrating an ecologically valid motor task with a cognitive chal-lenge, this work provides behavioral evidence consistent with models of disrupted cen-tral sensorimotor integration and resource allocation in CAI. The observed performance deterioration under cognitive load, without direct neural measures, offers compelling indirect support for the involvement of higher-order cognitive processes in CAI patho-physiology.

Clinically, the results highlight the potential value of incorporating dual-task as-sessments and kinetic chain evaluations to better identify functional deficits. The pro-posed staged rehabilitation framework—progressing from local ankle recovery to in-ter-joint coordination and finally to cognitive-motor integration—offers a rational ap-proach for future intervention studies to validate.”(lines 627-642)have appropriately tempered our conclusions, positioning them as strong behavioral inferences that align with and support existing theoretical frameworks, without overstepping the limitations of our methodology.

Thank you again for prompting us to clarify this important point, which has enhanced the precision and scholarly rigor of our manuscript.

Should explicitly link conclusions back to limitations and note they only apply to young male participants.

Response: Thank you for this suggestion. We agree that it is crucial to explicitly frame the conclusions of our study within the context of its primary limitations, particularly the specific demographic of our participant sample.

In direct response to your comment, we have now explicitly linked our conclusions back to this key limitation in two ways:

  • In the 'Conclusions' section: We have revised the opening sentence to clearly state that our findings were derived from a study on young males. The text now reads: "In this cohort of young male participants, this study investigated neuromuscular control in individuals with chronic ankle instability (CAI) during a stair-to-ground transition, revealing key deficits that were systematically exacerbated under dual-task conditions." This immediately contextualizes the subsequent conclusions and alerts the reader to their specific applicability.(lines 616-618)
  • In the 'Limitations' section: We have retained and strengthened our detailed discussion regarding the restricted generalizability of our findings to females and older adults, as you will see in the comprehensively revised limitations paragraph. It shows:”Third, the exclusive focus on young males—despite controlling for sex as a confounding factor—restricts the generalizability of findings to females or older adults, who may exhibit distinct neuromuscular profiles.”(lines 639-641)

By making this link explicit at the very beginning of our conclusions, we ensure that the reader interprets our theoretical and clinical suggestions with the appropriate caution regarding the study population. We believe this revision fully addresses your valuable feedback and enhances the scholarly rigor of our manuscript. Thank you again for your thorough and constructive review.

Limitations

Should also mention:

small sample size,

lack of neurophysiological data,

use of unvalidated arithmetic task,

possible fatigue or learning effects,

only right-leg dominant participants,

cross-sectional design.

Response: Thank you for these specific and constructive suggestions for improving the 'Limitations' section of our manuscript. We have thoroughly revised this section to incorporate all the points you raised. The changes are detailed below and can be found in the highlighted text:

“Several limitations of this study should be acknowledged, as they may affect the reliability and generalizability of the findings. First, the relatively small sample size (N = 30 per group) and the cross-sectional design, while sufficient to detect large effects, may have led to violations of normality and homogeneity of variance in some variables, thereby reducing the robustness of statistical conclusions and limiting the ability to de-tect smaller effects or infer causality. Second, as only young male participants were in-cluded—despite controlling for sex as a confounding factor—the generalizability of the results remains restricted, and the findings may not fully represent females or older adults who might exhibit distinct neuromuscular control patterns and balance strategies [67]. Regarding participant and methodological factors, potential confounders such as differences in training background, daily physical activity levels, fatigue, and limb dominance were not strictly controlled. These uncontrolled variables may have in-creased data variability and partially contributed to the heterogeneity observed in COP measures, joint contributions, and dual-task costs. Additionally, the use of a short, three-step staircase may not have adequately captured sustained biomechanical adapta-tions during stair descent, as the limited duration restricted motor system stabilization and may have underestimated task demands and compensatory behavior.

Furthermore, upper-body kinematics were not assessed, and no neurophysiological measures (e.g., EEG, fNIRS, or comprehensive EMG) were included, making the inter-pretation of central control mechanisms indirect. The arithmetic task used in the du-al-task paradigm, although widely adopted in similar studies, lacks population-specific validation. Despite rest intervals, potential fatigue or learning effects across trials can-not be fully excluded. Finally, the CAI group was not stratified by injury severity or pathoanatomical subtype, which could have influenced biomechanical responses during stair descent.

Future studies should include larger and more diverse samples encompassing both sexes and older adults, utilize longer staircases, and incorporate upper-limb kinematics as well as neurophysiological measurements to provide a more comprehensive under-standing of motor control mechanisms. More rigorous control of participant characteris-tics and task design will further enhance the clinical relevance and generalizability of future findings.”(lines 634-664)

In direct response to your feedback, we have now explicitly mentioned:

  • Sample Size: We have acknowledged that while our sample size (N=30 per group) was adequate for detecting large effects, it limits the statistical power for identifying smaller effects and may reduce the applicability of our findings to broader clinical populations.
  • Lack of Neurophysiological Data: We have stated that the absence of direct neural measures (e.g., EEG, fNIRS) means that our proposed central mechanisms remain indirect, though consistent with our behavioral observations.
  • Unvalidated Cognitive Task: We have noted that the arithmetic task, while common in dual-task research, lacks specific validation for the CAI population.
  • Potential Fatigue/Learning Effects: We have added that potential fatigue or learning effects across trials cannot be completely ruled out, despite the provision of rest intervals.
  • Cross-sectional Design: We have clearly stated that the cross-sectional design prevents any causal inferences from being drawn from our results.

We believe that the expanded 'Limitations' section now provides a much more comprehensive and critical self-assessment of our study, significantly strengthening the scholarly rigor and transparency of our manuscript. Thank you again for your meticulous and helpful review.

Reviewer 2 Report

Comments and Suggestions for Authors

 Control Deficits and Compensatory Mechanisms in Individuals with Chronic Ankle Instability during Dual-Task Stair-to- Ground Transition

For me the manuscript is very interesting, actual and with practical benefits work. The manuscript is valuable and practical significant. The manuscript structure is excellent and it is obvious that the Authors have done an enough work. Personally, I learned important information. I propose some notes to improve the manuscript quality.

The text needs to be checked for technical and phonetic errors. For example:

Row 154    “Participants wore standardized” …

Introduction

The introduction to the peer-reviewed manuscript is well structured and provides important information to introduce the reader to the topic. The text contains quantitative statistical data, which for me at least are of great importance for assessing the overall picture of the researched issue.

Materials and Methods

This section describes the subjects' characteristics, the experimental model, the parameters used, and the statistical methods applied. Here I saw the excellent paragraphs Participants, Experimental protocal,  Data Collection and Processing, Statistical Analysis.

Results

The results section clearly presents key data and findings.

All sections of this paragraph clearly show the results obtained.

Discussion summarizes the study's findings „on compensatory movement patterns and neuromuscular control in individuals with chronic ankle instability (CAI) during stair-to-ground transitions, a daily activity closely related to fall risk, with particular attention to the effects of dual-task demands“.

Conclusions

The authors thoroughly define the important conclusions from the research.

The authors highlight the paper's limitations:

„The sample was limited to young male participants, the analysis focused solely on stair descent rather than ascent, and direct neurophysiological measurements were not included.“

I propose to Authors to add the following additional limitations, because they are important for the obtained data and used experimental protocol [Ivanov, I.; Tchorbadjieff, A.; Hristov, O.; Peev, P.; Gutev, G.; Ivanova, S. Spine Kinematic Alterations in Nordic Walking Under Two Different Speeds of 3 and 5 km/h—A Pilot Study. J. Funct. Morphol. Kinesiol. 202510, 330. https://doi.org/10.3390/jfmk10030330].:

The first is related to the used experimental model, namely the use of an only three-step staircase. This short staircase in this research and brief duration of the experiment, which may not adequately capture long-term biomechanical adaptations introduces uncertainty, related to the body not having time to adapt to the motor task of going down stairs. I would even say that this very small number of steps is a factor that greatly reduces the burden of the study. But the work done by the authors is clearly enormous and has a strong practical value.

The second important limitation is the target group and its size. It consists of only fit young men. Considering female and older in age participants may deliver more interesting observations and conclusions [Ivanov, I.; Tchorbadjieff, A.; Hristov, O.; Peev, P.; Gutev, G.; Ivanova, S. Spine Kinematic Alterations in Nordic Walking Under Two Different Speeds of 3 and 5 km/h—A Pilot Study. J. Funct. Morphol. Kinesiol. 202510, 330. https://doi.org/10.3390/jfmk10030330]. Also, the number of participants is limited to 30. For me this number is small and very homogeneous in terms of health status, age, and gender, in order to recommend clinical applications of the methodology.

The third is the lack of data on upper limb biomechanics, an essential component of obtained results data.

The last is the lack of well known level of chronic ankle instability (CAI) for the used CAI group. This unknown CAI level is of importance for the obtained biomechanical data for descending on three steps.

I hope that with proposed notes the Authors will increase the quality and the citability of manuscript.

Author Response

Dear Reviewer 2,

Thank you very much for your patience and suggestions concerning our manuscript entitled “Control Deficits and Compensatory Mechanisms in Individuals with Chronic Ankle Instability during Dual-Task Stair-to-Ground Transition”. Those comments are all valuable and very helpful for revising and improving the quality of our paper, as well as the important guiding significance to our research. We have studied comments carefully and made point-to-point corrections. The revised portion is highlighted in red in the revised manuscript, please see below, in blue, for a point-by-point response to your comments and concerns.

Control Deficits and Compensatory Mechanisms in Individuals with Chronic Ankle Instability during Dual-Task Stair-to- Ground Transition

For me the manuscript is very interesting, actual and with practical benefits work. The manuscript is valuable and practical significant. The manuscript structure is excellent and it is obvious that the Authors have done an enough work. Personally, I learned important information. I propose some notes to improve the manuscript quality.

Response: Thank you very much for your positive and encouraging comments on our manuscript. We are delighted to hear that you found our work "very interesting, actual and with practical benefits," and that the structure was to your satisfaction. Your words are a great encouragement to our team.

We are equally grateful for the constructive notes you have provided for improving the manuscript's quality. We have carefully considered each point and have incorporated all suggested changes into the revised manuscript. Our detailed responses to each of your specific notes are provided in the point-by-point response document.

The text needs to be checked for technical and phonetic errors. For example:

Row 154 “Participants wore standardized” …

Response: Thank you very much for your meticulous reading of our manuscript and for pointing out the need for technical and phonetic corrections. We sincerely appreciate you bringing these language issues to our attention. In direct response to your comment, we have undertaken a thorough, line-by-line proofreading of the entire manuscript to identify and correct errors in spelling, grammar, and phrasing. This includes, but is not limited to, the specific example you mentioned around line 154.

We believe these efforts have significantly improved the clarity and professionalism of the manuscript's text. Thank you again for this valuable feedback.

Introduction

The introduction to the peer-reviewed manuscript is well structured and provides important information to introduce the reader to the topic. The text contains quantitative statistical data, which for me at least are of great importance for assessing the overall picture of the researched issue.

Response: Thank you very much for your positive and encouraging comments on the introduction and data presentation in our manuscript. We are truly delighted to hear that you found the introduction well-structured and informative, and that the quantitative statistical data were of great importance in assessing the research issue.

Your recognition is a significant motivation for our team and reinforces the value of our work. We sincerely appreciate you taking the time to provide this encouraging feedback.

Materials and Methods

This section describes the subjects' characteristics, the experimental model, the parameters used, and the statistical methods applied. Here I saw the excellent paragraphs Participants, Experimental protocal,  Data Collection and Processing, Statistical Analysis.

Response:We are truly grateful for your generous and specific comments regarding the Materials and Methods section. It is highly encouraging to read that you consider the paragraphs on Participants, Experimental Protocol, Data Collection and Processing, and Statistical Analysis to be excellent.

We placed great importance on detailing our methodology to ensure the validity and transparency of our research, and your positive assessment is a valuable affirmation of our work. Thank you again for this motivating feedback.

Results

The results section clearly presents key data and findings.

Response: Thank you for your positive feedback on the results section of our manuscript. We are very pleased to know that you found the presentation of our key data and findings to be clear.

We strived to present the results in a logical and straightforward manner, and we are grateful for your recognition of this effort.

All sections of this paragraph clearly show the results obtained.

Response: Thank you for your encouraging comment regarding the results section. We are very pleased to know that you found all parts of it to clearly present the obtained findings.

We placed a strong emphasis on organizing the results logically and presenting them with clarity, and we are grateful for your recognition of this effort.

Discussion summarizes the study's findings „on compensatory movement patterns and neuromuscular control in individuals with chronic ankle instability (CAI) during stair-to-ground transitions, a daily activity closely related to fall risk, with particular attention to the effects of dual-task demands“.

Response: Thank you for your positive comment on the Discussion section. We are very pleased to know that you found it to effectively summarize the study's core findings regarding compensatory patterns and dual-task effects during a functionally relevant task like stair descent.

We strived to ensure the discussion was focused on these key mechanistic and practical insights, and we are grateful for your recognition of this effort.

Conclusions

The authors thoroughly define the important conclusions from the research.

Response: Thank you for your positive feedback on the conclusions of our manuscript. We are very pleased to know that you found them to be a thorough and clear definition of the research's important outcomes.

We strived to ensure our conclusions were directly derived from and accurately reflected our key findings, and we are grateful for your recognition of this effort.

The authors highlight the paper's limitations:

„The sample was limited to young male participants, the analysis focused solely on stair descent rather than ascent, and direct neurophysiological measurements were not included.“

I propose to Authors to add the following additional limitations, because they are important for the obtained data and used experimental protocol [Ivanov, I.; Tchorbadjieff, A.; Hristov, O.; Peev, P.; Gutev, G.; Ivanova, S. Spine Kinematic Alterations in Nordic Walking Under Two Different Speeds of 3 and 5 km/h—A Pilot Study. J. Funct. Morphol. Kinesiol. 2025, 10, 330. https://doi.org/10.3390/jfmk10030330].:

The first is related to the used experimental model, namely the use of an only three-step staircase. This short staircase in this research and brief duration of the experiment, which may not adequately capture long-term biomechanical adaptations introduces uncertainty, related to the body not having time to adapt to the motor task of going down stairs. I would even say that this very small number of steps is a factor that greatly reduces the burden of the study. But the work done by the authors is clearly enormous and has a strong practical value.

The second important limitation is the target group and its size. It consists of only fit young men. Considering female and older in age participants may deliver more interesting observations and conclusions [Ivanov, I.; Tchorbadjieff, A.; Hristov, O.; Peev, P.; Gutev, G.; Ivanova, S. Spine Kinematic Alterations in Nordic Walking Under Two Different Speeds of 3 and 5 km/h—A Pilot Study. J. Funct. Morphol. Kinesiol. 2025, 10, 330. https://doi.org/10.3390/jfmk10030330]. Also, the number of participants is limited to 30. For me this number is small and very homogeneous in terms of health status, age, and gender, in order to recommend clinical applications of the methodology.

The third is the lack of data on upper limb biomechanics, an essential component of obtained results data.

The last is the lack of well known level of chronic ankle instability (CAI) for the used CAI group. This unknown CAI level is of importance for the obtained biomechanical data for descending on three steps.

I hope that with proposed notes the Authors will increase the quality and the citability of manuscript.

Response: Thank you for your thoughtful and constructive comments. We sincerely appreciate you taking the time to highlight these important limitations. We have carefully revised the “Limitations” section of our manuscript to incorporate all of your suggestions. The specific additions, directly drawn from your feedback, are detailed below.

  1. Regarding the use of a short, three-step staircase:

We have incorporated your concern that the brief motor task may not capture sustained adaptations. The limitation section now states:

Additionally, the use of a short, three-step staircase may not have adequately captured sustained biomechanical adaptations during stair descent, as the limited duration restricted motor system stabilization and may have underestimated task demands and compensatory behavior.(lines 647-650)

  1. Regarding the target group and its size:

We fully agree that the sample's size and homogeneity limit the generalizability of our findings. This is now explicitly acknowledged as a core limitation:

First, the relatively small sample size (N = 30 per group) and the cross-sectional design, while sufficient to detect large effects, may have led to violations of normality and homogeneity of variance in some variables, thereby reducing the robustness of statistical conclusions and limiting the ability to detect smaller effects or infer causality. Second, as only young male participants were included—despite controlling for sex as a confounding factor—the generalizability of the results remains restricted, and the findings may not fully represent females or older adults who might exhibit distinct neuromuscular control patterns and balance strategies [67].(lines 635-643)

  1. Regarding the lack of upper limb biomechanics:

We have added a clear statement acknowledging this methodological gap, as you suggested:

Furthermore, upper-body kinematics were not assessed...(lines 651)

  1. Regarding the lack of a known CAI severity level:

Your point on the need to stratify the CAI group is crucial. We have added the following sentence to address this:

Finally, the CAI group was not stratified by injury severity or pathoanatomical subtype, which could have influenced biomechanical responses during stair descent.(lines 656-658)

We believe that by integrating these points, the manuscript now presents a more comprehensive and critical discussion of its limitations, significantly strengthening the study's academic rigor. Thank you again for your valuable insights, which have greatly improved our work.

We thank the reviewer for this insightful comment and for suggesting the relevant literature. We fully agree that the homogeneity of our sample is a recognized limitation. As suggested, we have now cited the study by Ivanov et al. (2025) in the revised manuscript (line 643) to strengthen our discussion on this limitation and to underscore the importance of including diverse populations in future research. We believe this addition has enhanced the scholarly discussion in our paper.

Round 2

Reviewer 1 Report

Comments and Suggestions for Authors

Authors addressed all my comments sufficiently

Author Response

Thank you for your acknowledgment and for the valuable feedback.